



# Assessment of horizontally-oriented ice crystals with a combination of multiangle polarization lidar and cloud Doppler radar

Zhaolong Wu[1, 2], Patric Seifert[2], Yun He[2, 3, 4], Holger Baars[2], Haoran Li[5, 6], Cristofer Jimenez[2], Chengcai Li[1], and Albert Ansmann[2]

[1]Department of Atmospheric and Oceanic Sciences, School of Physics, Peking University, Beijing 100871, China
[2]Leibniz Institute for Tropospheric Research, Leipzig 04318, Germany
[3]School of Electronic Information, Wuhan University, Wuhan 430072, China
[4]State Observatory for Atmospheric Remote Sensing, Wuhan 430072, China
[5]Leipzig Institute for Meteorology (LIM), Leipzig University, Leipzig 04103, Germany
[6]State Key Laboratory of Severe Weather, Chinese Academy of Meteorological Sciences, Beijing 100081, China

**Correspondence:** Chengcai Li (ccli@pku.edu.cn)

**Abstract.** The orientation of ice crystals plays a significant role in determining their radiative and precipitating effects, horizontally oriented ice crystals (HOICs) reflect up to ~40 % more short-wave radiation back to space than randomly oriented ice crystals (ROICs). This study for the first time introduces an automatic pixel-by-pixel algorithm for HOIC identification using a combination of ground-based zenith- and 15-degree off-zenith-pointing polarization lidars. The lidar observations provided

high-resolution cloud phase information. The data were collected in Beijing over 354 days in 2022. A case study from 13 October 2022 is presented to demonstrate the effectiveness and the feasibility of the detection method. The synergy of lidars and collocated Ka-band cloud radar, radiosonde, and ERA5 data provide phenomenological insights into HOIC events. While cloud radar Doppler velocity data allowed the estimation of ice crystal size, Reynolds numbers, and turbulent eddy dissipation rates, corresponding environmental and radar-detected variables are also provided. HOICs were present accompanying with

weak horizontal wind of 0–20 $\mathrm{m\,s^{-1}}$ and relatively high temperature between −8 °C to −22 °C. Compared to the ROICs, HOICs exhibited larger reflectivity, spectral width, turbulent eddy dissipation rate, and a median Doppler velocity of about 0.8 $\mathrm{m\,s^{-1}}$. Ice crystal diameter (1029 μm to 1756 μm for 5th and 95th percentiles) and Reynolds numbers (28 to 88 for 5th and 95th percentiles) are also estimated with the help of cloud radar Doppler velocity using an aerodynamic model. One interesting finding is that the previously found switch-off region of the specular reflection in the region of cloud base shows a higher tur-

bulence eddy dissipation rate, probably caused by the latent heat released due to the sublimation of ice crystals in cloud-base region. The newly derived properties of HOICs have the potential to aid to derive the likelihood of their occurrence in output from general circulation models (GCMs) of the atmosphere.

## 1 Introduction

It has been recognized in the presence of Reynolds number between 1 and 100 (Pruppacher and Klett, 1996) that falling ice

crystals in the atmosphere can become quasi-horizontally oriented, only slightly deviating from the horizontal alignment due to wobbling movements. Frequently observed atmospheric optical phenomena (halos) including sun dogs (parhelia), light pillars



(sun pillar, moon pillar), circumzenithal arcs, and circumhorizontal arcs, etc. require the presence of horizontally oriented hexagonal plates. Tangent arcs require horizontally oriented hexagonal columns (Liou and Yang, 2016; Saito and Yang, 2019). Both crystal types, in general described as horizontally oriented ice crystals (HOICs) can produce angle-dependent specular
reflection for the incident light. The effect of that specular reflection defines the cloud radiative properties over large areas. In fact, regions with dominant HOICs can produce remarkable sunlight glints with much higher reflectance than the surroundings, as was observed by low-Earth orbit (Bréon and Dubrulle, 2004) and deep-space passive satellites (Marshak et al., 2017; Várnai et al., 2019; Li et al., 2019b).

HOICs reflect more shortwave radiation into space compared to randomly oriented ice crystals (ROICs), up to 40% more
according to modeling studies (Takano and Liou, 1989), thereby significantly influencing the radiation balance (Klotzsche and Macke, 2006). Mie scattering calculation shows oriented plates intercept roughly twice as much sunlight as the perfectly randomly oriented ones (Várnai et al., 2019). Stillwell et al. (2019) confirmed the significant radiation difference for HOIC and ROIC using a long-term ground-based dataset. Additionally, horizontal orientation increases the drag force from the atmosphere and thus slows the sedimentation speed of ice crystals, increasing the cloud lifetime and persistence in atmospheric models
(Heymsfield and Iaquinta, 2000).

Mirror-like specular reflection also strongly influences lidar observations. When the incident light is perpendicular to the main facets of HOIC, very strong backscatter and nearly no depolarization (specular reflection at normal incidence does not rotate the plane of polarized light) are found for zenith-pointing lidar (nadir-pointing for spaceborne lidar case). However, when the incident light is several degrees off perpendicular to the surface of HOIC, a relatively weaker backscatter, and higher
depolarization ratio is found for off-zenith-pointing lidar (off-nadir-pointing for spaceborne lidar case). This angle-dependent characteristic is beneficial for distinguishing the HOICs from ROICs (He et al., 2021a, b; Seifert, 2011). As another crucial feature, HOICs can lead to misclassification of the cloud phase based on the zenith/nadir polarization lidar-based cloud phase discrimination due to the similarity of the near-zero depolarization ratios produced by both, specular reflection at HOICs and backscattering from droplets of supercooled water cloud. To avoid specular reflection from HOICs, spaceborne lidars posi-
tioned several degrees off-nadir to capture the cloud phase information better, 3° for ATLID (Atmospheric Lidar) onboard EarthCARE (Earth Cloud, Aerosol and Radiation Explorer) and CALIOP (Cloud-Aerosol Lidar and Infrared Pathfinder Satellite Observations), and 2° for ACDL (Aerosol and Carbon dioxide Detection Lidar) onboard DQ-1 (Wehr et al., 2023; Hu et al., 2009; Dai et al., 2024) respectively. Many ground-based lidars and ceilometers were also positioned several degrees off-zenith to reduce the HOIC contamination of supercooled liquid droplet identification schemes (e.g., Engelmann et al., 2016).

Despite its importance, limited knowledge exists regarding HOICs. Due to the perturbation of ice orientation by the detector, it is very difficult to use airborne in-situ methods to measure the ice orientation. Remote sensing methods, including both ground-based measurement and spaceborne observations were developed and employed to investigate the characteristics of HOICs. Diattenuation, a polarization-dependent measure of scattering efficiency shown by oriented particles at so-called oblique angles, was proposed by Neely et al. (2013) to study HOICs and their radiation effects in Greenland (Stillwell et al.,
2019). Seifert (2011) used zenith-pointing polarization Raman lidar to retrieve the lidar ratio of HOICs and pointed out that HOICs show a lower lidar ratio than supercooled water clouds. Westbrook et al. (2010) used the ratio of backscatter (color



ratio) from an off-zenith pointing ceilometer and a zenith-pointing Doppler lidar to identify and study the HOICs, though his study lacked depolarization ratio capacity. He et al. (2021a), with a 30 ° off-zenith lidar along with a zenith-pointing lidar, used the enhanced volume depolarization from off-zenith-pointing lidar as a feature to identify HOICs, and found the horizontal ori-
entation can form from a continuously descending ice cloud layer. However, they didn't use the backscatter as a restraint and his identification method was manual. CALIOP separated HOIC from ROIC and liquid water cloud based on layer-integrated attenuated backscatter and depolarization ratio threshold (Hu et al., 2009). With its global coverage, CALIOP shows a pow- erful advantage in observing the global distribution of HOICs. However, lidar attenuation when a liquid-water-topped cloud exists could lead to an underestimation of the HOICs fraction. The relatively coarse spatial resolution and layer-integrated,
vertically homogeneous (within a determined cloud layer) official cloud phase classification (Hu et al., 2009) is not detailed enough to investigate the horizontal orientation. The spaceborne lidar is more suitable for global-scale statistics, providing only a snapshot observation, which cannot observe the process-level evolution of HOIC. Passive satellite using glint to identify HOICs can provide the macroscopic distribution of oriented ice (Marshak et al., 2017; Bréon and Dubrulle, 2004), but without height-resolved information.

Fundamental questions such as about the frequency of HOIC still persist, literature provides different results as a conse- quence of the variability of the underlying detection and counting methods: profile-based (Ross et al., 2017), pixel-based (Westbrook et al., 2010), cloud-layer-based (Zhou et al., 2012a), or area-based (Bréon and Dubrulle, 2004; Marshak et al., 2017). Westbrook et al. (2010) points out that many of the results from different studies are inconsistent. Noel and Chepfer (2010) found 6% optically thin ice cloud contains oriented ice using nadir-pointing CALIPSO data. Zhou et al. (2012b) esti-
mated HOIC exists in approximately 60% of optically thick ice and mixed-phase cloud layers. Marshak et al. (2017) pointed out that roughly every third DSCOVR/EPIC image (see http://epic.gsfc.nasa.gov) shows a sunglint over land, which is most likely due to HOICs. The automated algorithm proposed in this paper can serve as a good starting point for future HOIC frequency and percentage studies.

More observations with HOIC-identification capabilities are needed to improve the assignment of the orientation of ice
hydrometeors in the cloud parameterization schemes, which is usually not considered in current general circulation models or radiative transfer models (Klotzsche and Macke, 2006; Zhou et al., 2012b). Due to the latitude-dependent HOIC occurrence (Noel and Chepfer, 2010), the long-term observation at midlatitude stations like Beijing (116.3°E, 40.0°N), China is essential to help understand the orientation phenomenon.

Previous ground-based statistics are mostly case studies describing areas with specular reflection effects, lacking precise
height-resolved pixel-level observations and products. Compared to spaceborne observations, the ground-based dual-angle- lidar scheme has a higher spatial and temporal resolution to analyze the evolution of HOICs, which is beneficial to our under- standing of the process. It gives us a more comprehensive understanding of the environmental characteristics of the emergence of HOICs. In addition, simultaneous cloud radar observations, which very few HOIC studies utilized so far, can obtain Doppler velocities, which help us to estimate ice crystal size information, and to derive turbulence information which helps to under-
stand under which conditions HOICs tend to form.



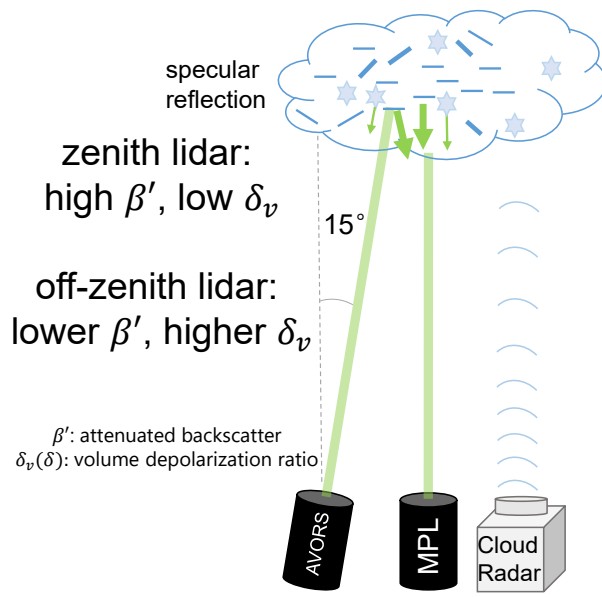

**Figure 1.** Schematic figure of the two lidars and the zenith-pointing cloud radar, which were used in the framework of this study.

For the first time, the simultaneous observations using ground-based zenith and 15 degrees off-zenith-pointing polarization lidars along with cloud radar (see Fig. 1) were conducted to study HOIC in Beijing, China over 354 days in the year 2022. This article proposes an automatic algorithm for HOIC detection based on dual-angle polarization lidar observations and then explores the potential of such a unique system using collocated cloud radar, radiosonde and ERA5 data. The article is structured as follows. Section 2 presents the instruments and data used. In Sect. 3, we detail the methodology employed to detect the cloud pixels and then identify HOICs. Next, a case study is shown in Sect. 4. Finally, conclusions are drawn in Sect. 5.

## 2 Instruments and data

We propose a new and unique set of instruments to study HOIC occurence and formation: 2 polarization lidars with different zenith angles, Doppler cloud radar. The depolarization ratio measured by zenith-pointing MPL and 15° off-zenith-pointing AVORS lidar was calibrated using well-calibrated Raman lidar as a reference, as is described in details in Appendix B2.

### 2.1 MPL

A Micro Pulse Lidar (MPL; Model MPL-4B-IDS-532-AT) manufactured by Sigma Space Corporation has been continuously operated since 2016 (Chu et al., 2019; Xu et al., 2020; Chang et al., 2021), at zenith angle on the roof of the Peking University (PKU) Physics Building (116.3°E, 40.0°N, ~40 m above sea level). The temporal and vertical resolution of the MPL are 15 s and 15 m, respectively, and the blind zone is 195 m (Welton and Campbell, 2002). The pulse repetition frequency is 2500 Hz,



and the pulse energy is 6–8 $\mu$J. The MPL is a single-wavelength (532 nm) elastic polarization lidar with a field of view (FOV) of 0.1 mrad. By using the actively controlled liquid crystal retarder (LCR), the MPL achieves polarization detection capabilities by using only one detector for the two polarization channels (Flynna et al., 2007). The MPL was placed in a container with air conditioning to maintain stable observations. On top of the container, there is a slightly tilted glass to guarantee that lidar observations are not interrupted by bad weather. A lens hood above the glass reduces the sunlight noise. Maintenance staff carefully wipe the glass above the lidar container every day to minimize attenuation caused by rain and aerosol deposits. Unless otherwise stated, the time in the paper is local time (LT, UTC+8).

## 2.2 AVORS lidar

An off-zenith-pointing lidar (Model Portable Particle Lidar, http://en.avorstech.com/product/670.html) manufactured by AVORS Technology has been continuously operating since 2022 (Sun et al., 2024). The AVORS lidar was placed 5 meters away from the MPL on the same roof. The AVORS lidar is a single-wavelength (532 nm) elastic polarization lidar, with a rotatable base and an electric motor to change its zenith and azimuth angle. The laser beam of the AVORS lidar was placed 15 degrees off the zenith (towards the north to reduce the possible sunlight noise to the greatest extent) to avoid specular reflection during our study. The pulse energy is 20 $\mu$J with the pulse repetition frequency of 2500 Hz, and the FOV of the telescope is 0.2 mrad. The temporal and vertical resolution of the AVORS lidar data are 10 s or 60 s (adjustable, most of the time 60 s in this research) and 15 m respectively, and the blind zone is 45 m. Technical specifications of the lidar system are listed in Table A1. Note, the AVORS lidar has two Photomultiplier Tube (PMT) detectors for each polarization channel (Fig. A1). In the following analysis, the height of observations from the AVORS lidar of was calculated as $\cos(75°)$ times the range from the lidar. The AVORS lidar was installed outdoors and the temperature was maintained by its own air conditioning system integrated inside the lidar. Due to the lack of containers or lens hoods, the lidar signal is slightly more contaminated by sunlight noise in the daytime, resulting in a relatively lower signal-to-noise ratio (SNR). Furthermore, the AVORS lidar was zenith-pointing from 9 May 2023 to 3 June 2023, during this period the depolarization ratio for cloud could be compared with MPL (Fig. B2).

The uniqueness of the 15° off-zenith angle observation is valuable in this research. Previous research shows that the 3° off-nadir angle of CALIOP was not sufficient to completely eliminate the effects of specular reflections (Noel and Chepfer, 2010; Kikuchi et al., 2021), hence CALIOP possesses the ability to offer a product about oriented ice at the 3° off-nadir angle. Also, ground-based polarization lidar with 4° or 5° off-zenith-pointing angle can also sometimes show specular reflections (Tansey et al., 2023; Seifert et al., 2011). The 30° off-zenith angle of He et al. (2021a) used is too large, and there will be a large horizontal offset at high altitude. In this study, a 15° off-zenith angle is a moderate angle, while avoiding the backscatter specular reflection of HOICs as much as possible, and also trying to ensure that the same cloud can be seen by both lidars.

## 2.3 Raman lidar

Portable, eye-safe lidars, such as MPL and AVORS, have polarization capabilities. Preliminary results revealed that the calibration of the systems needs to be improved in order to make the collocated measurements comparable. A well-characterized lidar was used as a reference to characterize the two micropulse lidar systems. A Raymetrics Raman lidar (Model LR231-D300) at





the same campus (about 360 m away from MPL and AVORS lidar) was employed as the reference for the depolarization ratio

(Li et al., 2016, 2019a; Tan et al., 2019, 2020a, b; Ren et al., 2021). The Raman lidar operates at three wavelengths (355, 532, and 1064 nm), two of them 355, 532 nm are equipped with polarization channels. In the present study, the 532 nm channel of this Raman lidar has been designated as a reference for the calibration of the depolarization ratios of the other two lidars (see B2 in detail). Its performance of depolarization measurement has been verified by several previous studies (Tan et al., 2020a, b). The $\Delta 90^{\circ}$ method was employed to ensure its accuracy for depolarization ratio (Freudenthaler et al., 2009). Note

that we do not use this Raman lidar for HOIC identification due to its discontinuous observation.

## 2.4 Cloud radar

To further explore the potential of the new approach in the investigation of HOIC events, additional information from a radar instrument was considered. The larger wavelength of the radar instrument makes it able to penetrate deeper into clouds compared to lidar, it is also more sensitive to large hydrometeors. The Doppler spectrum provides an estimation of the falling

velocity of particles, which allows us to estimate the particle size using aerodynamic models. The radar measurements were also used to derive turbulence-related information, such as turbulent eddy dissipation ratio (EDR) and Reynolds number (see Appendix D and E), which may play a significant role in the orientation of ice crystals. To our best knowledge, no Doppler cloud radar data are used to investigate the identified horizontally oriented ice except for Westbrook et al. (2010) and Stillwell et al. (2018), so it provides a unique chance to investigate more radar-based characteristics for different orientation behaviors

of ice crystals.

A 33.44 GHz Ka-band solid-state, depolarization, multi-mode, zenith-pointing millimeter-wave cloud radar (MMCR, Model HMB-KP) manufactured by Beijing Institute of Radio Measurement has been continuously operating at Peking University since 2018 (Wang et al., 2022; Zhang et al., 2024). The cloud radar is placed 5 meters beside the MPL container. The temporal resolution is about 13 s (adjustable), and the vertical resolution is 30 m. The radar operates in four alternating modes: boundary

layer mode (mode 1), cirrus mode (mode 2), precipitation mode (mode 3), and middle-level mode (mode 4) (Ding et al., 2022). These four modes vary in pulse compression ratios and numbers of both coherent and incoherent integrations. The boundary layer mode is designed to identify low-altitude clouds by utilizing a narrower pulse waveform and increasing the number of coherent integrations to enhance detection capability. In the cirrus mode, pulse compression techniques are employed to boost sensitivity for detecting high-altitude clouds with weaker radar echoes. The precipitation mode features an extended

unambiguous range and velocity measurements tailored for observing rainfall. The middle-level mode similarly applies pulse compression techniques but with a reduced number of coherent integrations. Furthermore, there is one combined mode (mode 8) that combines all the modes to produce one final observation result, which we use in this research. The radar measurements contain raw data of Doppler spectra and spectral moments including reflectivity, mean Doppler velocity, spectrum width, and linear depolarization ratio.



## 2.5   Radiosonde and ECMWF ERA 5 reanalysis data

Radiosondes were launched every day at 00:00 UTC (08:00 local time) and 12:00 UTC (20:00 local time) at the Beijing Nanjiao meteorological site (116.47°E, 39.80°N, WMO NO. 54511), 25 km from our lidar site (Chu et al., 2019), providing meteorological parameters, e.g., temperature, relative humidity, and horizontal wind speed and direction. As a measure to compensate for the time sparsity of the radiosonde, ECMWF ERA5 reanalysis from the grid point of PKU (116.3°E, 40.0°N) was used to provide the meteorological parameters, i.e., the temperature, wind, and relative humidity over ice. ERA5 reanalysis data for 2022 were compared with simultaneous radiosonde profiles, as in Yin et al. (2021)., and the difference in temperature, wind speed, and relative humidity are $0.2 \pm 0.86$ °C, $0.53 \pm 1.98$ m s$^{-1}$, and $-5.46 \pm 12.69$ %, respectively, indicating reliability of ERA5 data for our analysis.

## 3   Methodology

This section introduces a new identification scheme and describes the analysis procedure. First, raw lidar data were calibrated to obtain attenuated backscatter coefficient (see Appendix B1). Second, lidar, cloud radar, and ERA5 data were re-gridded (averaged or interpolated) to 5 min× 15m resolution. Then the following algorithms and corrections were applied to get the HOIC and other hydrometeor types.

### 3.1   Cloud layer identification algorithm

An advanced value distribution equalization method (Zhao et al., 2014) was applied to identify cloud pixels from lidar backscatter signals. Next, the overlap region of the cloud pixels detected by both MPL and AVORS lidars was selected for the further cloud phase determination algorithm. For HOIC cases, zenith-pointing lidar observations have a much stronger backscatter than off-zenith-pointing lidar observations. The identification of cloud pixels mainly depends on the backscatter, the cloud detection algorithm can identify one pixel as a cloud at least when the backscatter of the pixels reaches a predefined threshold whose determination is explained by Zhao et al. (2014). The limitation of the cloud detection algorithm to those regions where the signals of both lidars overlap will lead to an underestimation of some upmost HOIC pixels (beyond the off-zenith lidar attenuation region but still clear on zenith lidar observation, compare Figs. 4a, c, and g).

It is essential to evaluate whether the two lidars detect the same cloud layer by estimating the typical horizontal deviation of the two laser beams at cloud height. For a cloud at an altitude of 6 km, the horizontal deviation from the zenith-pointing lidar is 6 km $\times \tan(15°) = 1.6$ km. Assuming an horizontal wind speed is $v = 20$ m s$^{-1}$ (see radiosonde Fig. 6b), the horizontal movement of the cloud is 6000 m within five minutes, which is the temporal resolution utilized in data processing. Consequently, if both lidars observe the same cloud within the same time slot (> 5 min), the horizontal deviation of the off-zenith pointing lidar is less significant (1.6 km < 6 km). Although with increasing height, the horizontal distance between the probed volumes also increases (from 0.268 km at 1 km height to 2.68 km at 10 km height.).





## 3.2 Cloud phase determination algorithm

A specialized algorithm (Fig. 2) is applied to differentiate between different cloud phases for the intersecting cloud pixels observed by both lidars. First, we utilized the temperature of homogeneous nucleation ($<-38°C$) to distinguish the ice phase, followed by using the off-zenith-pointing lidar volume depolarization ratio of 0.1–0.3 to identify ice-containing cloud pixels at temperatures between 0 and $-38°C$. Cloud pixels with a volume depolarization ratio of $\geq 0.3$ are categorized as randomly oriented ice crystals (ROICs). Cloud pixels with a volume depolarization ratio of 0.1–0.3 are categorized as mixed-phase cloud pixels (MPCs). If by contrast, the above $\delta_{\text{off-zenith}} > 0.1$ ice-containing cloud pixels show $\delta_{\text{zenith}} < 0.1$, then we used the zenith-to-off-zenith ratio of attenuated backscatter > 2 and the zenith-to-off-zenith ratio of volume depolarization ratio < 0.6 as stringent criteria to exclusively capture the most representative signals of HOIC, i.e. the specular reflection effect is strong enough without ambiguity. On the contrary, ice-containing pixels that do not meet the three thresholds are kept their ROIC or MPC labels. It should be mentioned that the real case of the orientation of the ice crystals is always a mixture within a lidar-detected bulk (Saito and Yang, 2019; Borovoi et al., 2018). This means that the HOIC label indicates the specific pixel contains HOIC (with a certain proportion) which produces an unambiguous specular reflection signal; however, some ROICs may still exist in the pixel.

The threshold values were fixed empirically from the whole cloud dataset collected during 2022. The criteria used to select a typical HOIC pixel are shown in Fig. 3. Figure 3a is the median volume depolarization ratio as a function of temperature for all detected cloud layers in 2022. The depolarization ratio $\delta$ indicates particle sphericity: ROICs show a higher volume depolarization ratio on the order of 0.3–0.5, while spherical liquid droplets have near-zero values (Ansmann et al., 2008; Seifert, 2011). A high depolarization ratio (> 0.3) at temperatures below $-38\,°C$ (the threshold temperature for homogeneous freezing) and a low depolarization ratio (< 0.1) at temperatures above $0\,°C$ are identified as the depolarization ratio criteria for ice and liquid water clouds, respectively. The two depolarization ratio threshold values are shown in red dashed line in Fig. 3a. Zenith lidar shows much lower depolarization than off-zenith lidar between $-40\,°C$ and $0\,°C$, probably due to contamination of the data by HOICs. Figure 3b shows the ratio of attenuated backscatter and depolarization ratio by means of a density scatter plot which is based on all cloud pixels detected in 2022. Most cloud pixels accumulate around the 1:1 line, indicating that zenith and off-zenith lidar have comparable volume depolarization ratio and attenuated backscatter. In contrast, a distinct tail-like cluster is evident in the upper-left region of Fig. 3b. The lower depolarization and higher backscatter in zenith lidar observations compared to off-zenith lidar observations shown in this cluster are clear features of HOICs. For the identification scheme, we use the zenith-to-off-zenith ratio of the attenuated backscatter ratio greater than 2 and depolarization ratio less than 0.6 as criteria to avoid the most frequent cloud pixels (green, yellow, and red) in the center part of Fig. 3b. This is a relatively strict criterion as it considers both the criterion for attenuated backscatter and depolarization ratio to categorize the most representative pixels for specular reflection. In the case of liquid-water clouds, the zenith-to-off-zenith depolarization ratio differs from 1, ranging between 0.55 and 1 according to (Jimenez et al., 2020) as the ratio of lidar FOVs is 1:2. However, the backscatter ratio will be less than one, making the distinction of HOICs unambiguous with the described criteria.





Figure 3c shows the density scatter plot between off-zenith lidar's attenuated backscatter and depolarization ratio. Two evident clusters can be found: low depolarization ratio and high backscatter indicating liquid water clouds and high depolarization

ratio and low backscatter indicating ice clouds, respectively. Nonspherical ice crystals exhibit higher $\delta_v$ compared to spherical liquid water droplets, whereas droplets have higher $\beta'$ due to their higher concentration. The black dashed lines ($\delta_v < 0.1$ and $\beta' > 5 \, \mathrm{Mm^{-1}sr^{-1}}$) indicate the criteria for the classification of liquid water. If a cloud pixel meets the two introduced criteria and has a temperature $\geq 0 \, °C$, it is categorized as liquid water (or water as an abbreviation). If the temperature is below $0 \, °C$, it is then flagged as a supercooled liquid water cloud (SWC). Note that if a cloud pixel meets a relatively low depolarization ratio

($\delta_v < 0.1$) but does not meet the attenuated backscatter criterion ($\beta' \leq 5 \, \mathrm{Mm^{-1}sr^{-1}}$), it is classified as non-typed cloud pixels (similar to Baars et al., 2017). This usually happens when the concentration of cloud particles is low or the pixels are actually some dense aerosol particles (e.g. mineral dust). Discriminating optically-thin clouds and dense aerosol is still a challenge for a cloud mask algorithm based on lidar backscatter signal, thus we exclude the non-typed cloud pixels from further analysis in this study.

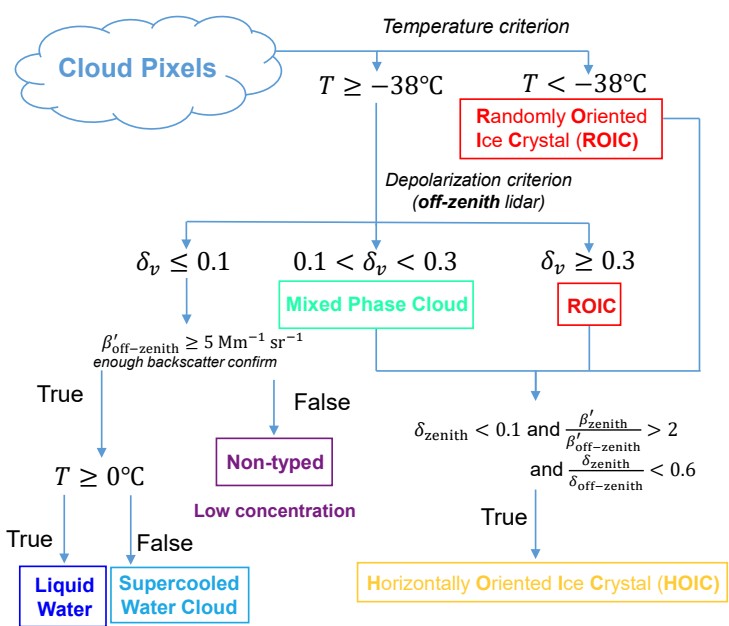

**Figure 2.** Flowchart of the algorithm for the classification of hydrometeor types.

## 3.3 Cloud phase correction

After applying the cloud phase discrimination algorithm, the potential multiple scattering effects and the contamination of the molecular depolarization ratio for ice clouds have to be considered, thus the following two corrections were conducted.



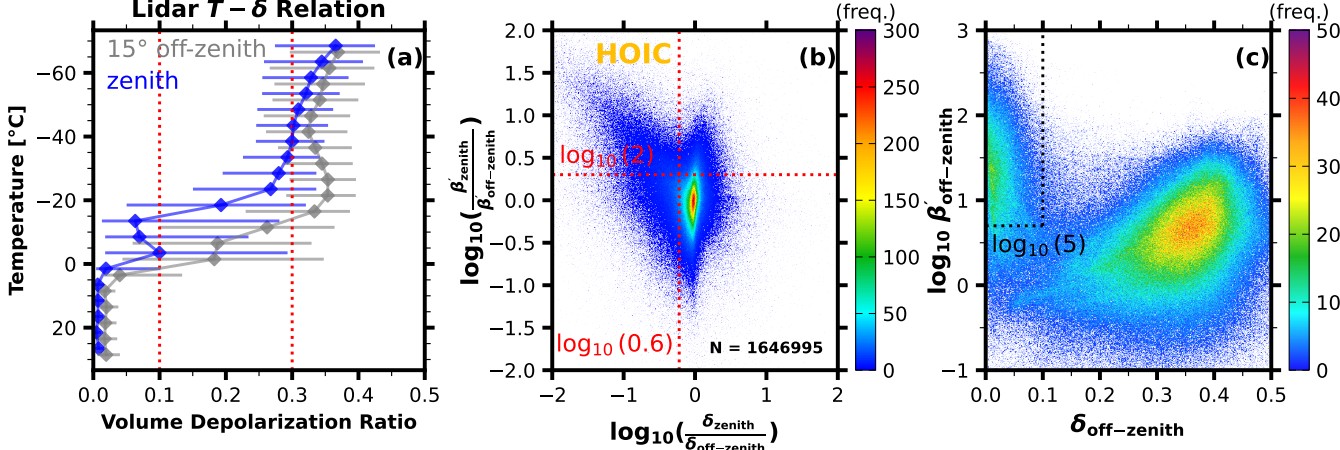

**Figure 3.** Definition of criteria for identification of the HOICs (red dashed lines). All subfigures were created using all the cloud pixels observed in 2022. **(a)** Median volume depolarization ratio as a function of temperature for each height bin within all detected cloud pixels in temperature increments of 5 °C. Horizontal bars indicate the interquartile range (IQR). **(b)** Density scatter plot of the ratio of zenith- to off-zenith-pointing lidar's volume depolarization and attenuated backscatter, $\log_{10}$ scale. **(c)** Density scatter plot of volume depolarization ratio and attenuated backscatter (unit in $Mm^{-1}sr^{-1}$, $\log_{10}$ scale) for off-zenith-pointing lidar. Black dashed lines indicate the criteria used for the identification of liquid water.

### 3.3.1 Typing correction above liquid layers

The off-zenith lidar, with its greater FOV (0.2 mrad compared to 0.1 mrad), generally results in a higher depolarization ratio
at the top of water clouds than the zenith-pointing MPL (see Fig. 3a, T > 0°C region). This occurs when the effect of multiple scattering is pronounced as it penetrates the water cloud. Due to the different deadtime reactions for the two lidar detectors, MPL shows stronger attenuated backscatter at low altitudes (< 1.2 km). We thus categorized HOIC pixels below 1.2 km as liquid water cloud (T ≥ 0 °C) or supercooled liquid water cloud (T < 0 °C). For water clouds above 1.2 km, if the topmost 100 meters of a profile have been flagged as HOIC but the layer shows liquid water towards the bottom, the upper pixels will
be classified as liquid water. This criteria allows us to exclude possible artifacts led by horizontal homogeneities and strong attenuation of the laser beam in the liquid cloud layer. In this way, this method conservatively corrects most misclassified HOICs at the top of water clouds back to the water phase.

### 3.3.2 Ice virga correction beneath pure ice clouds

As the elastic backscatter lidars we used for our study lack Raman or high spectral resolution channels, we use the volume
depolarization ratio instead of the particle depolarization ratio, to avoid the introduction of additional uncertainties by the assumption of lidar ratios in the Klett-Fernald method (Fernald, 1984) that would be required for calculation of the ratio of particle to molecular backscattering. Application of the volume depolarization ratio however introduces certain ambiguities





within thin clouds or in presence of low concentrations of ice crystals, as it is the case, e.g., in the ice virga region beneath pure ice clouds. The reason is, that the magnitude of the volume depolarization ratio depends on the relative contribution of the molecular depolarization ratio which is caused by backscattering at air molecules (Cairo et al., 1999). With the decreasing contribution of particle backscattering, the molecular backscattering and the associated low depolarization ratio of 0.004 (in the case of AVORS lidar) contaminates the total signal and decreases the effective volume depolarization ratio of the ice crystals, resulting in the ice pixel not reaching the depolarization-ratio criteria of 0.3 for the off-zenith-pointing AVORS lidar. This leads to them being categorized improperly as mixed-phase cloud pixels, even though actually no water droplet may exist under this circumstance. For MPC pixels near the bottom of a cloud, the following procedure was thus applied. If more than 5 out of the 10 adjacent cloud pixels above an MPC cloud pixel contain ROICs, and the temperature is below $-20\,°C$, they are re-categorized as ROICs. The threshold of $-20\,°C$ is a typical criterion for the sharp decrease of the fraction of supercooled liquid water (Yorks et al., 2011; Wang et al., 2019), even though a small likelihood for the presence of liquid water persists down to $-38\,°C$ (see, e.g., Radenz et al. (2021) and Fig. 4g), 20:00 to 21:00 at heights above the $-22\,°C$ isotherm.

## 4 Case study on 13 October 2022

Figure 4 illustrates a case study of a mid-level cloud layer from 13 October 2022, when strong specular reflections were observed with the zenith-pointing lidar for almost the whole day. A high backscatter and low depolarization ratio in the zenith-pointing lidar observations and much lower backscatter and higher depolarization in the off-zenith-pointing lidar measurements indicated the presence of HOICs. The HOIC flag as derived by the algorithm introduced in this study is denoted in orange in Fig. 4g. Figure 5 shows average profiles of selected lidar and radar parameters for the period from 11:00 to 12:00 (local time) on 13 October 2022. The attenuated backscatter coefficient of HOIC (yellow shaded region) observed by zenith-pointing lidar is nearly two orders of magnitude larger than that observed by the off-zenith-pointing lidar (Fig. 5a). The volume depolarization ratio from the zenith-pointing lidar is nearly zero, while for the off-zenith lidar observation, the peak volume depolarization ratio exceeds 0.3 (Fig. 5b). Additionally, Fig. 5c and d, respectively, illustrate that the radar reflectivity factor is greater for HOIC while Doppler velocity is smaller, compared to ROIC.

This HOIC event persisted for nearly the whole day. Some HOIC pixels showed strong attenuated backscatter for the zenith-pointing lidar but relatively lower attenuated backscatter for the off-zenith-pointing lidar. Still, between 00:00 and 03:00 such pixels were not identified as cloud, probably because the backscatter observed by the off-zenith-pointing lidar was too small for triggering the cloud mask detection for the off-zenith-pointing lidar. This demonstrates the stringency of the cloud identification criterion of the HOIC detection algorithm.

### 4.1 General description of HOIC event

It is noteworthy that zenith lidar observations for some regions show a high depolarization ratio below the cloud levels of strong specular reflection. Prominent time-height regions where this was the case, are for instance the time periods from 03:00–08:00 and from 11:00–15:00 at the height level of 4–5 km. This phenomenon is described as the 'switch-off' of the specular reflection





**Figure 4.** Lidar ((**a**)-(**g**)) and zenith-pointing Ka-band cloud radar ((**h**)-(**l**)) observations on 13 October 2022, time-height contour plots (5 min / 15 m resolution for (**a**)-(**g**), 13 s / 30 m resolution for (**h**)(**i**) to show the variation of Doppler velocity, 5 min / 30 m for (**j**)-(**l**)). (**a**) 15 ° off-zenith-pointing lidar attenuated backscatter. (**b**) 15 ° off-zenith-pointing lidar volume depolarization ratio. (**c**) Zenith-pointing lidar attenuated backscatter. (**d**) Zenith-pointing lidar volume depolarization ratio. (**e**) The ratio of attenuated backscatter for zenith-pointing and off-zenith-pointing lidar. (**f**) The ratio of volume depolarization ratio for zenith-pointing and off-zenith-pointing lidar. (**g**) Cloud phase categorization results with isotherm from ERA 5 data. Abbreviations of SWC, ROIC, HOIC, and MPC represent supercooled liquid water cloud, randomly oriented ice crystal, horizontally oriented ice crystal, and mixed-phased cloud. There is no cloud pixel categorized as (warm) water due to the subzero temperature. (**h**)(**i**)(**k**)(**l**) Cloud radar detected momentum data: Doppler velocity, spectral width, reflectivity (with isotherm from ERA 5 data), and linear depolarization ratio (LDR). (**j**) Cloud radar retrieved eddy dissipation rate (EDR, $\epsilon$).



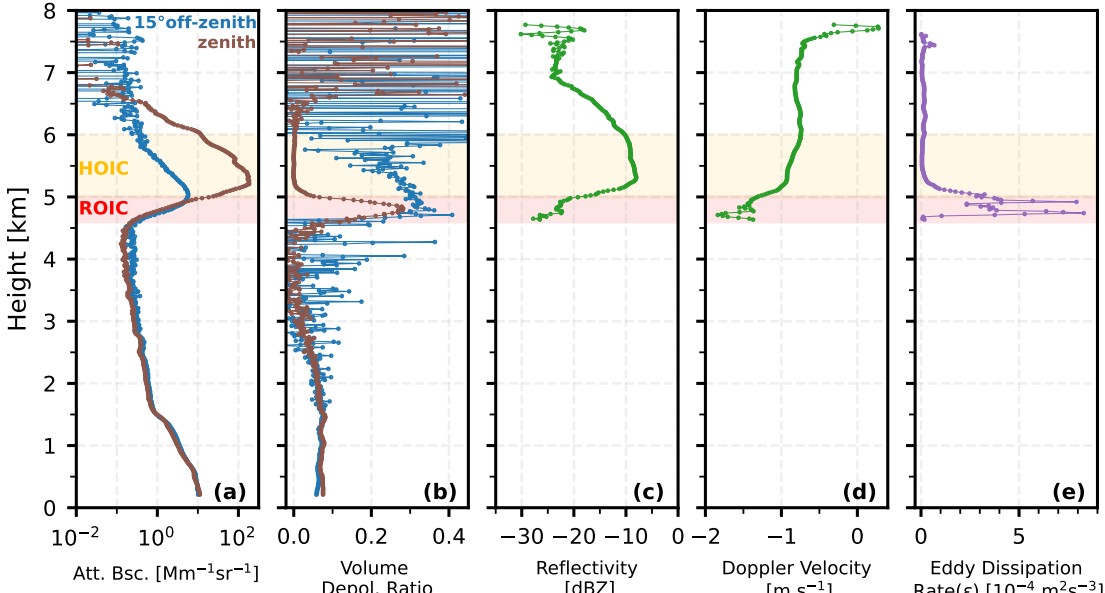

**Figure 5.** Lidar **(a-b)** and radar **(c-d)** observations at 11:00-12:00 on 13 October 2022; **(a)** shows the attenuated backscatter profiles from both zenith and off-zenith pointing lidars. **(b)** shows the volume depolarization ratio from both lidars. **(c)** and **(d)** show the radar reflectivity and Doppler velocity, respectively.**(e)** Retrieved eddy dissipation rate. The shadowed orange regions denote the presence of HOICs, and the red regions represent the dominance of ROICs.

conditions (Westbrook et al., 2010; He et al., 2021a). Having the observations of both depolarization lidars and Doppler cloud radar available, this study aims to zoom in on this phenomenon in more detail than was done in previous studies. The cloud radar observations (Fig. 4) show that the Doppler velocity changes significantly with time at the cloud base of the high-depolarization regions at the cloud base (compare Figs. 4d and 4h). The zenith lidar observation's high depolarization ratio region also coincides with a relatively higher spectral width of Doppler velocity (Fig. 4i). A higher spectral width usually means stronger turbulence, more complex particle spectral distributions, stronger wind shear, beam broadening within the region, etc (Kollias et al., 2007). In order to separate the effect of turbulence from other factors affecting spectral width, the turbulent eddy dissipation rate (EDR, $\epsilon$) was computed to reflect the turbulence (Figs. 4j and 5e) using quantities including the standard deviation of Doppler velocity and horizontal wind speed. Details on the EDR retrieval are outlined in Appendix E. Measuring the turbulent kinetic energy (TKE) dissipation rate, or turbulent eddy dissipation rate, which represents the rate of conversion of TKE into heat or the rate at which the TKE is dissipated by viscosity, is a good way to estimate the turbulence activity. As a quantitative proxy of atmospheric turbulence, a large EDR indicates rapid energy dissipation and high atmospheric turbulence (Griesche et al., 2020). The high $\delta_v$ region above the cloud base (Fig. 5b, brown line within red shaded region, also see Fig. 4d) in the zenith lidar observation has a higher eddy dissipation rate (Fig. 5e, also see Fig. 4j), suggesting the strong turbulence caused by the latent heat released due to the sublimation of ice crystals near the cloud base. Another possible explanation is



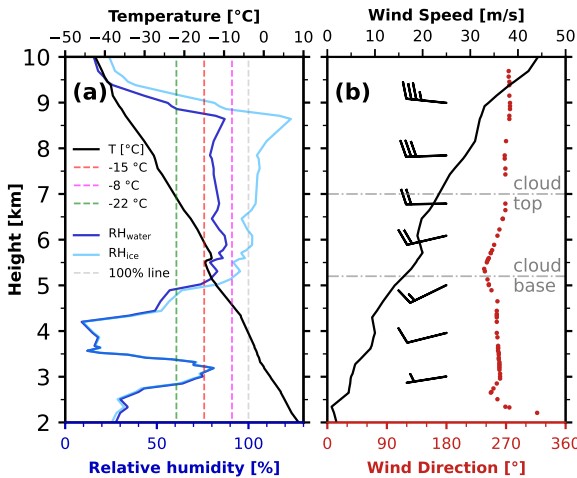

**Figure 6.** Beijing Nanjiao radiosonde profile (25 km away from PKU) at local time (UTC+8) 08:00 on 13 October 2022. **(a)** Temperature (black) and relative humidity with respect to liquid (deep blue) and ice (light blue) profile; **(b)** Wind speed (black), wind direction (red), and wind barbs. Two gray horizontal dash-dotted lines represent the cloud base, approximately 5.2 km, and the cloud top, around 7 km, respectively.

that this stronger turbulence in return causes the break up of the ice crystal orientation. Horizontally oriented ice crystals need calm dynamic conditions and low turbulence to maintain their quasi-horizontal orientation (Klett, 1995; Garrett et al., 2015).

The following subsections analyze the environment variable, cloud radar observed variables, and diameter and Reynolds numbers retrieved for HOICs. Then, the relationship between supercooled water clouds and the different orientations of ice crystals is discussed.

## 4.2 Environment variables

Figure 6 shows the measurements of a radiosonde that was launched at Beijing Nanjiao station at 08:00 LT on 13 October, including the temperature and relative humidity, and the horizontal wind speed and direction profile. The temperatures of $-8\,°C$, $-22\,°C$ and $-15\,°C$ are denoted using magenta, green and light red dashed lines, respectively. Between temperatures of $-8\,°C$ and $-22\,°C$ plate-like ice crystals tend to form, according to the ice crystal habit diagram, i.e. morphology of ice crystals as a function of temperature (Libbrecht, 2005; Li, 2021; Bailey and Hallett, 2009). Overall, the highest probability for the occurrence of HOIC was reported to occur at the temperature level of $-15\,°C$ (Westbrook et al., 2010). In Fig. 6a, the temperature for typical specular reflection (between 5 and 7 km) is around $-15\,°C$, which falls within the plate-like ice crystal temperature range of $-22\,°C$ to $-8\,°C$, while the relative humidity over ice (Fig. 6a, light blue line) approaches or slightly exceeds 100%. As seen from Fig. 6b, the wind was western and relatively light (approximately 10–20 $m\,s^{-1}$) at the altitudes of the HOIC layer, which is beneficial for HOIC to maintain. Since the horizontal orientation is quasi-horizontal with some fluttering or wobbling angle, an increase in horizontal wind speed may impinge upon the principal facet of the HOIC,





generating significant torque that could potentially disrupt their orientation. Figure 6b also shows the wind speed and direction of the cloud base region (around 5.2 km) changed sharply along the different altitudes. This wind shear could induce turbulence in this region, thereby establishing conditions conducive to disrupting the horizontal orientation of ice crystals.

Figure 7 shows the normalized frequency of HOIC and ROIC under different horizontal wind speed and temperature conditions. It can be concluded that HOIC usually occurs accompanied by smaller horizontal wind speeds and higher temperatures. Additionally, Figure 7c shows the density scatter plot of horizontal wind speed and temperature conditions where HOICs occur. High concentration (deep green) of HOIC pixels lies at higher temperatures ($-5$ °C to $-18$ °C) and lower horizontal wind speed (2 to 20 m s$^{-1}$). A negative correlation is found between horizontal wind speed and temperature where HOICs exist.

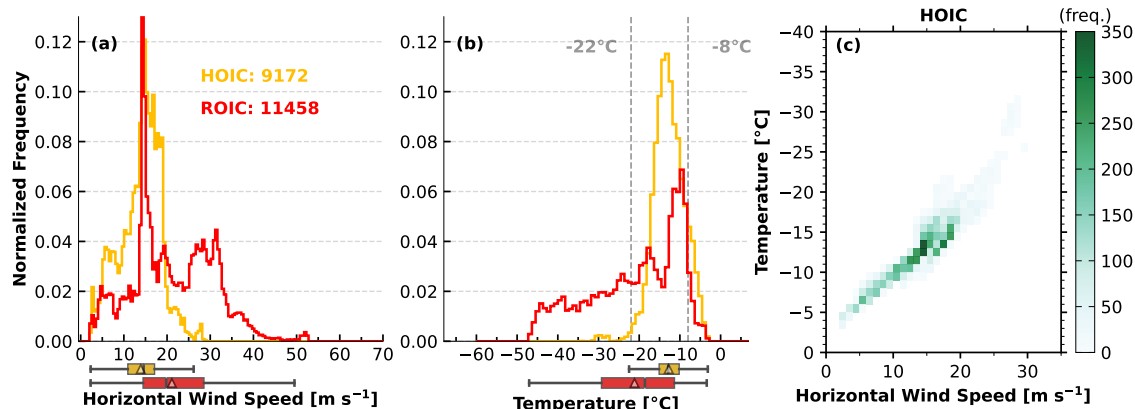

**Figure 7.** Frequency distributions of environmental variables of different cloud phase classes on 13 October 2022. **(a)** The normalized histogram of the horizontal wind speed where HOICs and ROICs exist, with boxplots shown below the x-axis. The boxes extend from the lower to upper quartile values, with grey lines at the median, and triangles at the mean. The whiskers extend from the box to the minimum-maximum values or extend from the box by 1.5 times the interquartile range. The flyers are not shown in the plot. **(b)** The normalized histogram of temperature where HOICs and ROICs exist, with boxplots shown below the x-axis. **(c)** The density scatter plot of horizontal wind speed and temperature where HOICs exist, the greener the color, the higher the number density of HOIC pixels.

**4.3    Cloud radar observations**

From cloud radar observations, we can obtain reflectivity, Doppler velocity, and spectral with for HOICs and ROICs as shown in Fig. 8. The Doppler velocity for HOICs is more narrow and shows a smaller maximum value than that for ROICs, which is consistent with the findings by (Westbrook et al., 2010). The median Doppler velocity of HOIC is approximately $-0.8$ m s$^{-1}$, and as Westbrook et al. (2010) shows from their Doppler lidar observation, the fall speeds for the oriented ice are about

$-0.3$ m s$^{-1}$. The longer wavelength of operation of cloud radar, compared to lidar, likely results in the measurement of higher Doppler velocities, as they are more sensitive to larger particles than lidar. Generally, larger particles tend to have higher fall velocities.



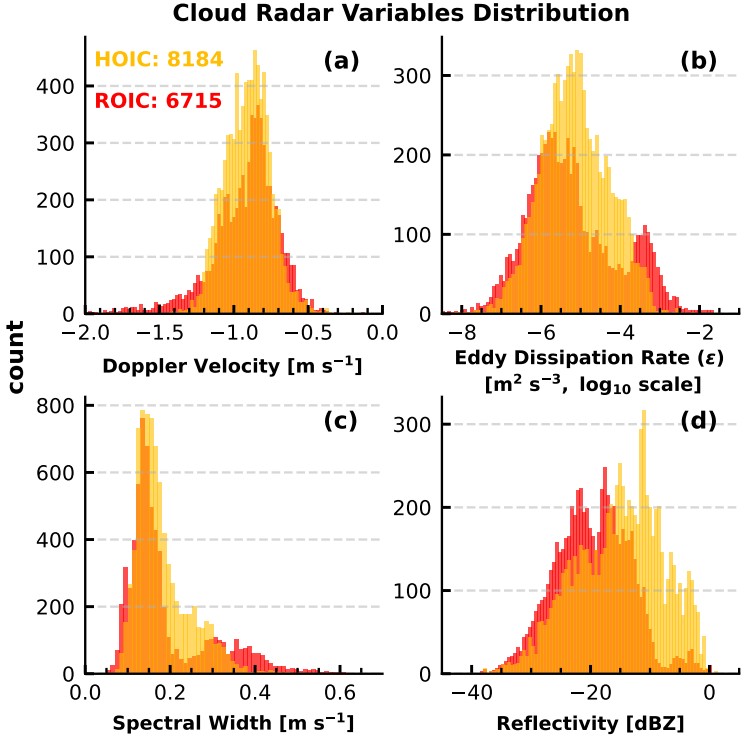

**Figure 8.** Cloud radar variables distributions of different cloud phases on 13 October 2022. Where HOICs and ROICs exist **(a)** the histogram of Doppler velocity. **(b)** the histogram of retrieved eddy dissipation rate. **(c)** the histogram of spectral width. **(d)** the histogram of reflectivity.

Figure 8(b) shows that the largest eddy dissipation rate ($> 10^{-3}$ m$^2$ s$^{-3}$) occurs only with ROICs. Low EDR occurs with both ROIC and HOIC, there are no significant differences between these two orientations. Fig. 8(c) shows that the largest spectral width corresponds to the presence of ROICs. The same behavior is found for the EDR (Fig. 8b). Bimodal structures are found for both EDR and spectral width.

The reflectivity shows larger values for HOICs than ROICs in this case. The peak of the HOICs (around $-10$ dBZ) is much higher than the peak of ROICs (around $-20$ dBZ), indicating that a larger $nD^6$ is detected in the bulk of the region where HOICs exist. Contrary to its sensitivity for the presence of columnar ice crystals (Li et al., 2021) or the melting layer (Li and Moisseev, 2020), the cloud radar LDR seems to be rather insensitive to HOIC as shown in Fig. 4(l).

## 4.4 Diameter and Reynolds number retrieved for HOICs

Figure 9 shows the diameter and Reynolds number of HOICs retrieved with an aerodynamic model by assuming that the shape of HOIC can be described by a hexagonal plate. The retrieval methods are described in Appendix D in detail. A summary of the statistical properties of estimated diameters and Reynolds numbers are listed in Table 1. The crystal diameters are mostly between 700 and 2000 μm, with a median of 1355 μm and a mean of 1367 μm, which is consistent with the values of 100–



3000 μm from Polarization and Directionality of the Earth Reflectances (POLDER) data (Bréon and Dubrulle, 2004, Fig. 10). Sassen (1980) revealed that a crystal diameter of > 100–200 μm is needed for maintaining the horizontal orientation using photographic analyses of light pillar displays. He et al. (2021a) reported estimated diameters of 464–1305 μm for HOICs in 12 cases, with associated Reynolds numbers of 4–58 using a profile-based approach.

Most of the retrieved Reynolds numbers are between 1 and 100 (see Fig. 9d), which coincides with the value of 0.39–80 from the estimation based on spaceborne passive satellite observation (Bréon and Dubrulle, 2004). This study used a pixel-based approach to obtain a wider range of Reynolds numbers than He et al. (2021a)'s falling profile-based approach. Kajikawa (1992) measured the lower critical values of the Reynolds number for unstable falling motion (random orientation) of ice crystals, resulting in 47 to 90.7 based on the different crystal shapes (47 for hexagonal plate).

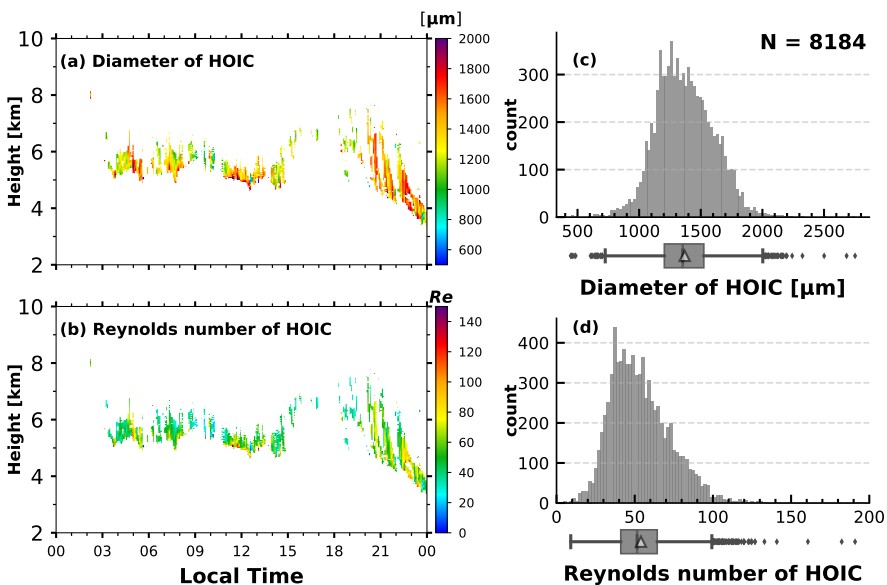

**Figure 9. (a)** Diameter and **(b)** Reynolds number of HOIC retrieved on 13 October 2022, time-height contour plots (5 min / 15 m resolution). And histogram of HOIC's retrieved diameter **(c)** and Reynolds number **(d)**, with the boxplots below the x-axis (the boxes extend from the lower to upper quartile values, with grey lines at the median, triangles at the mean. The whiskers extend from the box to the minimum-maximum values or extend from the box by 1.5 times the interquartile range).

**4.5   Relationship between supercooled water cloud and HOICs**

At heights above identified HOIC pixels, we can sometimes find some supercooled water pixels. As Westbrook et al. (2010) point out, supercooled water clouds likely play an important role in the formation of HOIC. Since we have a supercooled water cloud product (light blue in Fig. 4g and Fig.10), we can also prelimarily investigate the relationship between the occurrence of the supercooled water cloud class and HOICs based on the case study.





**Table 1.** Statistics of estimated diameter and Reynolds number for HOICs on 13 October 2022.

| Statistics | Diameter [μm] | Reynolds number |
|---|---|---|
| 5th percentile | 1029 | 28 |
| First quartile | 1204 | 39 |
| Median | 1354 | 51 |
| Third quartile | 1525 | 65 |
| 95th percentile | 1756 | 88 |
| Mean | 1369 | 54 |

Especially at times after 16:00, Fig. 4 indicates the supercooled water on top of the identified HOIC regions, similar as was discussed earlier by, e.g., Westbrook et al. (2010) and He et al. (2021a). Probably the pristine ice crystals which are generated in the supercooled water layers are more liable to maintain a horizontal orientation. In turn, aging and further processing of ice crystals by means of riming, aggregation, or breakup probably alters the ice crystal structure toward more complex shapes which are associated to a smaller torque to maintain the orientation. At times before 16:00 in Fig. 4, the scenario is more complex. For this period, a comparison to the cloud radar observation (Fig. 4k) indicates that the signals of both lidar systems were subject to strong attenuation. For most of the time, the cloud radar detected much higher cloud tops as was identified by the lidars. This was especially the case for those time periods where HOIC was identified. It is thus likely that (1) the HOIC were formed at higher altitudes/lower temperatures and (2) that a relationship to the existence of liquid water cannot be directly evaluated because no liquid layers could be identified due to the strong attenuation. Apart from this caveat, the temperatures of the radar-derived cloud tops provide strong hints that liquid water was also involved in the formation of the HOIC observed before 16:00. As noted in Fig. 4k, the top temperatures were generally above $-25\,°C$. It is known from previous studies (De Boer et al., 2011; Westbrook and Illingworth, 2011) that ice forms only via the liquid phase at temperatures above that threshold.

Taking the above discussed indications granted, it appears reasonable to evaluate the relationship between the occurrence of liquid water and HOIC in more detail in a quantitative way. In order to do so, we calculate the Euclidean distance between ROIC and HOIC, respectively, and the supercooled water cloud pixels. The relationship which was derived for the observations on 13 October 2022 is shown in Fig. 10. The Euclidean distance is derived by taking the square root of the sum of the squares of the horizontal and vertical distances. The horizontal distance was computed by multiplying the horizontal wind component from ERA5 by the time interval between the pixels, while the vertical distance is the height difference between the targeted pixels. Moreover, considering the inherent falling characteristics of ice crystals and the general increase of the wind velocity with height, we focus solely on the earlier (leftward in the time-height cross-section) and higher (upward in the time-height cross-section) supercooled water cloud pixels, as they potentially affect the alignment of the ice crystals. From Fig. 10a it can be seen that HOICs in comparison to ROICs are in general closer (brighter in color shade) to regions of supercooled liquid water.




Figure 10b illustrates the quantitative statistical analysis of the Euclidean distance between HOICs and supercooled water
pixels. From this Figure, it is evident that the Euclidean distance for HOICs relative to pixels of supercooled liquid water is
smaller than that for ROICs, with both a lower median and mean value. This indicates that HOICs are, in general, physically
closer to supercooled water clouds. Even considering potential lidar attenuation in this instance, it is still possible to prelim-
inarily conclude that supercooled water droplets may play a significant role in the formation of orientation. It is plausible

that pristine ice crystals, generated at cloud top temperatures between -8 °C and -22 °C, are more likely to induce horizontal
orientation with large facets to counteract drag force. Future research should encompass more extensive studies on this subject.

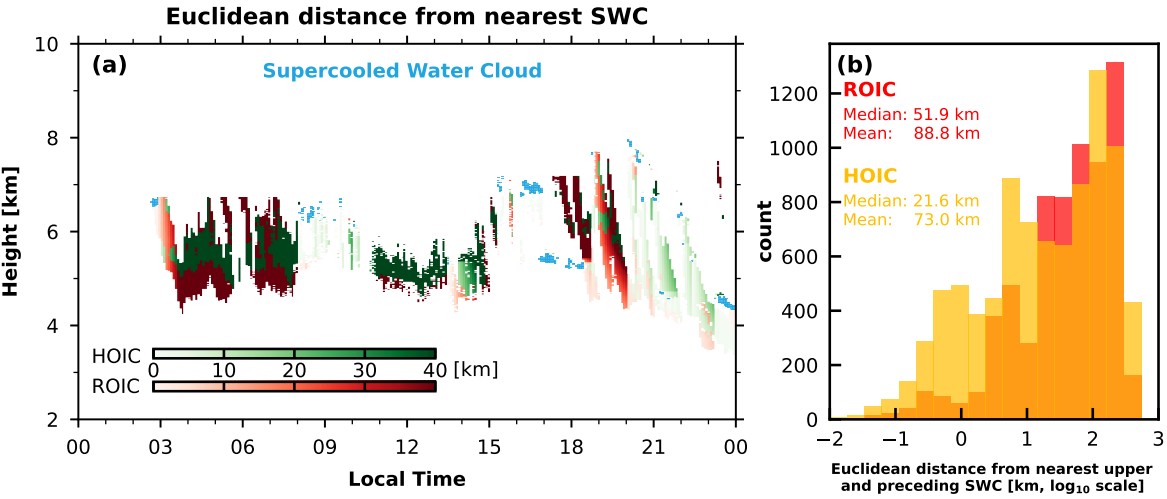

**Figure 10. (a)** The time-height cross-section of the Euclidean distance between the HOICs, ROICs, and supercooled water cloud pixels on
13 Oct. 2022. The light blue pixels represent the supercooled water cloud pixels. The color of the indicates the Euclidean distance, the darker
the color (dark green and dark ROIC), the farther the Euclidean distance. Note here we use a green colorbar instead of orange for HOICs
to better discriminate HOIC and ROIC with different lightness of color. **(b)** The histogram of the Euclidean distance between the HOICs,
ROICs, and supercooled water cloud pixels in $\log_{10}$ scale with the median and mean value noted in the upper left corner.

## 5    Summary and conclusions

In this study, we developed a novel pixel-by-pixel detection method for horizontally oriented ice crystals (HOICs) using
a combination of zenith-pointing and 15-degree off-zenith-pointing polarization lidars, in Beijing, China. In synergy with

collocated observations from cloud radar, radiosonde data, and the ERA5 dataset, our approach provides unprecedented detail
in HOIC detection and characterization of HOICs. This enhancement facilitates improvements in both the spatial and temporal
resolution of these observations, thereby enabling a comprehensive investigation into the phenomenological aspects of HOIC
events. One of the key findings of this research is the enhanced turbulence eddy dissipation rates (EDR) observed near the





cloud base, which corresponds to the "switch-off" phenomenon of horizontal orientation in ice crystals. We attribute this
phenomenon to the latent heat released from ice crystal sublimation. This discovery represents a significant advancement over
previous studies, providing new insights into the role of turbulence in disrupting the horizontal alignment of ice crystals.

Our case study showed that HOICs form in relatively warm temperatures ($-8\,°C$ to $-22\,°C$) where plate-like ice crystals
exist and the presence of rather light wind speeds ($0–20\,\mathrm{m\,s^{-1}}$). Cloud radar indicates that mean Doppler velocity is similar
for HOICs and ROICs (randomly oriented ice crystals), but more concentrated for HOICs. The highest EDR and spectral
width are exclusive to ROICs, while HOICs generally have larger reflectivity. Moreover, the estimated diameter using Doppler
cloud radar and the aerodynamic model (ranging from approximately 1029 μm to 1756 μm for 5th and 95th percentiles) and
Reynolds numbers (28 to 88 for 5th and 95th percentiles) provide a clearer understanding of HOICs' microphysical properties.
Moreover, our observations suggest a strong relationship between supercooled water clouds and HOIC formation, with a closer
Euclidean distance between supercooled water cloud pixels for HOICs than ROICs. The HOIC persisted for nearly the whole
day in this case, indicating the HOIC could significantly impact the radiation balance. These findings could help improve the
parameterization schemes in climate models, especially in mid-latitude regions like Beijing.

In this paper, we only show one typical case to demonstrate the HOIC identification method. More case studies could
be shown in the following work to show different conditions for HOICs (different cloud top temperatures). The observation
method and detection algorithm developed in this research provide new tools for long-term HOIC observation, due to the
precise pixel identification of HOICs and continuous observation, future work of diurnal and seasonal characteristics will be
established through the yearlong dataset. Since this dataset enables the joint classification of both supercooled water cloud
pixels and HOIC, it provides a unique dataset to investigate the relationship between supercooled water and HOIC, which
could shed light on the generation mechanism of HOICs, as previously revealed by Westbrook et al. (2010) and the Eucledian-
distance approach presented within our study. Except for the relationship between supercooled water and HOIC, even recent
studies suggest HOICs are often correlated in precipitating clouds and their ice nucleation processes have a connection with
precipitation formation (Ross et al., 2017; Kikuchi and Suzuki, 2019). In addition, the orientation of ice crystals is always a
mixture in a bulk volume, and the horizontal orientation is kind of quasi-horizontally oriented, with some flutter or wobbling
angles. The fraction of HOIC inside a pixel and the flutter angle could be retrieved with the help of HOICs model data
(Borovoi et al., 2018; Saito and Yang, 2019). Further studies are required help to derive the ratio of HOIC/ROIC. What's more,
in this study, we only use two elastic lidars, long-term Raman lidars and HSRLs observation could be employed to accurately
determine the lidar ratio and particle depolarization ratio to provide more information about HOICs in the future.

Like all the lidar-based research, lidar attenuation for opaque clouds (i.e. optical depths roughly above 3) is a main defect
for this method, so some upmost cloud pixels are missed. With cloud radar, we can infer the cloud top beyond lidar attenuation
to a certain extent. Future ice crystal orientation detection work based on radar should be carried out to make up for this defect
(Hajipour et al., 2024). The assumption of pure hexagonal plates in the diameter and Reynolds number retrieval is the simplest,
highly symmetrical crystal. In practice, many other planar crystals exist. Using Doppler velocity to estimate the ice crystal
diameter is a rough estimation because the superposition of air movement is ignored. Future retrieval could consider deriving
the ice crystal size utilizing the ratio of radar reflectivity and the lidar extinction (Bühl et al., 2019; Ansmann et al., 2024). So





some uncertainty could exist in the estimation of diameter and Reynolds number process. We only consider the middle part
of the retrieved data, without the extreme value. In conclusion, we find the collection of retrieval techniques and approaches
for HOIC classification and characterization, that was presented in this study, a valuable toolset for statistical evaluations that
cover larger time periods. This possibility is granted by means of the 1-year dataset from Beijing that was introduced only
briefly here. In a follow-up study, a statistical evaluation of the relationship between HOIC and other cloud microphysical and
environmental parameters will be presented.

## Appendix A:  AVORS lidar optomechanical setup

Figure A1 is the optomechanical setup of the AVORS lidar, an image courtesy of AVORS Technology. Note two photomultiplier
tubes (PMT, Model Hamamatsu H10682-210) are used in the system.

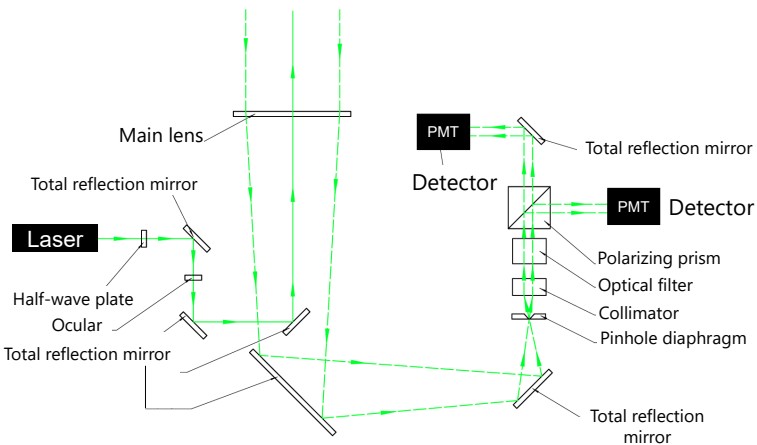

**Figure A1.** Schematic of the AVORS lidar optomechanical setup

## Appendix B:  Lidar calibration

### B1    Lidar system constant calibration

This step is from the lidar's original photon count number to lidar attenuated backscatter, to make the zenith-pointing and
off-zenith pointing lidars' backscatter signal comparable. First, for the MPL and AVORS lidar, both two lidars use a photon
count system. The photo count rate times dead time correction then minus the afterpulse value for MPL (no need for afterpulse
for AVORS lidar after the manufacturer's test), then minus the background, times the square of range and divided by overlap
function and laser energy. In this way, we get the normalized relative backscatter (NRB). Then we employ PollyNet's calibration
method (Baars et al., 2016; Yin et al., 2020) to two lidars' NRB profiles. Namely first Rayleigh fit, i.e. find the aerosol-free
region. Then we suppose a fixed lidar ratio of 50 sr, using the Klett-Fernald method to retrieve extinction profile (Fernald,



**Table A1.** Specifications of the Portable Particle Lidar by AVORS Technology

| Specifications | Value |
|---|---|
| **Transmitter** | |
| Laser wavelength (nm) | 532 |
| Laser energy ($\mu$J) | 20 |
| Pulse width (ns) | $\leq 10$ |
| Emission laser pulse linewidth (nm) | $\leq 0.2$ |
| Linear polarization purity | $> 100:1$ |
| Divergence angle (mrad) | $< 1$ |
| **Receiver** | |
| Diameter of narrow-FOV telescope (mm) | 160 |
| FOV of narrow-FOV telescope (mrad) | $< 0.2$ |
| **Detector and DAQ** | |
| Manufacturer of detector | Hamamatsu |
| Type of detector | PMT, H10682-210 |
| Photodetector mode | Photon counting |
| Sampling rate of DAQ (MHz) | $> 10$ |
| Sampling bits | Photon counting channel: 200 M c/s |
| Interference filter | Bandwidth: $\leq 0.2$ nm |
| | Out-of-band suppression $\geq$ OD4 |

1984). After that, we integrate the extinction profiles to get the height-resolved aerosol optical depth (AOD) and the lidar calibration constant, then we employ a smooth method to the lidar calibration constant profile to determine the final lidar calibration constant. After that, the attenuated backscatter can be derived by NRB divided by the lidar calibration constant.

## B2 Lidar depolarization calibration

Lidar depolarization is the key feature that distinguishes the HOIC. HOIC shows a much lower depolarization ratio than ROIC for zenith-pointing lidar (He et al., 2021a). Before the identification of HOIC, we must confirm the reliability of our lidars' depolarization ratio. We use the well-calibrated Raman lidar to get the depolarization ratio, and match the same time's MPL and AVORS lidar depolarization ratio.

Due to the MPL has only one detector for two polarization channels (Flynna et al., 2007), there is no gain ratio effect here (or gain ratio $K^* = 1$ in Papetta et al.'s case, see Eq. (B1)), we use the Córdoba-Jabonero et al. (2021) method to calibrate the MPL's depolarization, namely assuming a constant deviation compared with the reference Raman lidar. Figure B1a, b are the



scatter plots between the MPL depolarization ratios and reference Raman lidar depolarization ratios on 5 March 2022 from 00:00 to 06:00 and 14 April 2022 from 00:00 to 05:00, respectively, showing a constant intercept of $-0.02$. Then we simply use $\delta_{\mathrm{MPLcalibrated}} = \delta_{\mathrm{MPL}} - 0.02$ to calibrate the MPL depolarization. Figure B1c, d show the calibrated MPL depolarization ratio profiles (purple) and the observed uncalibrated ones (orange), indicating well agreement of depolarization ratio between the calibrated MPL and the reference Raman lidar. Note as it shows in Fig. B1c, for calibrated MPL depolarization ratio profile, there is a slight difference left from Raman lidar ($\approx 0.01$) below 3 km, but it is still acceptable.

For AVORS lidar, since it has two detectors for each polarization channel (similar to the Cimel CE376 lidar system used in Papetta's case, see Fig. A1), we use the newly proposed Papetta's two-parameter method to calibrate the depolarization of the lidars (Papetta et al., 2024, Eq. (10)). The two parameters are $K^*$, the gain ratio between the two channels, and $g$, the cross-talk from the co-polar signal to the cross-polar signal.

Using $\delta^*$ to denote uncalibrated AVORS lidar depolarization ratio, or observed depolarization ratio, the calibrated AVORS lidar depolarization ratio ($\delta$) can be expressed using $K^*$ and $g$:

$$\delta = \frac{\delta^*}{K^*} - g \tag{B1}$$

Then we select one dust layer and one aerosol-free region as reference, we use the $\delta_{\mathrm{ref}}^{\mathrm{d}}$ from the reference Raman lidar observation and theoretical molecular depolarization $\delta_{\mathrm{m}}$ 0.004 (Behrendt and Nakamura, 2002), and we select the observed lidar depolarization ratio of dust layer ($\delta_{\mathrm{d}}^*$) and the molecular layer ($\delta_{\mathrm{m}}^*$).

$$\delta_{\mathrm{ref}}^{\mathrm{d}} = \frac{\delta_{\mathrm{d}}^*}{K^*} - g \tag{B2}$$

$$\delta_{\mathrm{m}} = \frac{\delta_{\mathrm{m}}^*}{K^*} - g \tag{B3}$$

With two unknowns and two equations, the $K^*$ and $g$ could be solved using:

$$K^* = \frac{\delta_{\mathrm{d}}^* - \delta_{\mathrm{m}}^*}{\delta_{\mathrm{ref}}^{\mathrm{d}} - \delta_{\mathrm{m}}} \tag{B4}$$

$$g = \frac{\delta_{\mathrm{m}}^* \delta_{\mathrm{ref}}^{\mathrm{d}} - \delta_{\mathrm{d}}^* \delta_{\mathrm{m}}}{\delta_{\mathrm{d}}^* - \delta_{\mathrm{m}}^*} \tag{B5}$$

What's more, Fig. B1c shows the case (5 March 2022, 00:00-06:00) used for AVORS lidar depolarization calibration. Similar to Papetta's case, we select the molecule region around 7-7.5 km (Eq. (B2)) and dust region around 1 km (Eq. (B3)) as reference, both region was denoted with grey frame in Fig. 2c. After that we can get $K^* = 0.954$ and $g = 0.0329$ from Eqs. (B4) and (B5), and then the calibrated depolarization ratio profile from Eq. (B1). Figure B1c shows the calibrated AVORS lidar depolarization (black) and the observed one (pink). We can find that after calibration, the AVORS lidar depolarization ratio is very close to the reference Raman lidar (see Fig. 2c, back line, and blue line).



Figure B1d shows the calibrated MPL and AVORS lidar and the Raman lidar depolarization ratio profiles on 14 April 2022 from 00:00 to 05:00 (another case). For the molecule (above 11 km) and dust region (below 4km), the three lidars' depolarization ratio profiles match perfectly, indicating the calibration method used here for two lidars shows good performance. For the

cirrus region between 7.5 km and 11 km, however, there is a slight difference between the three lidars' depolarization ratio, which could be explained by the three lidars having different fields of view (Raman 2.3 mrad, MPL 0.1 mrad, AVORS 0.2 mrad), multiple scattering effects with the cloud could contribute some uncertainty. And since the AVORS lidar was off-zenith pointing (Raman lidar and MPL are both zenith-pointing), maybe some horizontal heterogeneity of the cirrus exists. Nevertheless, the slight difference is affordable for our algorithm 2 to distinguish HOIC. Note the Raman lidar depolarization matches

very well with theoretical molecular depolarization 0.004 (Fig. B1c, d above 7 km and 11km, respectively), showing excellent depolarization performance as a reference lidar.

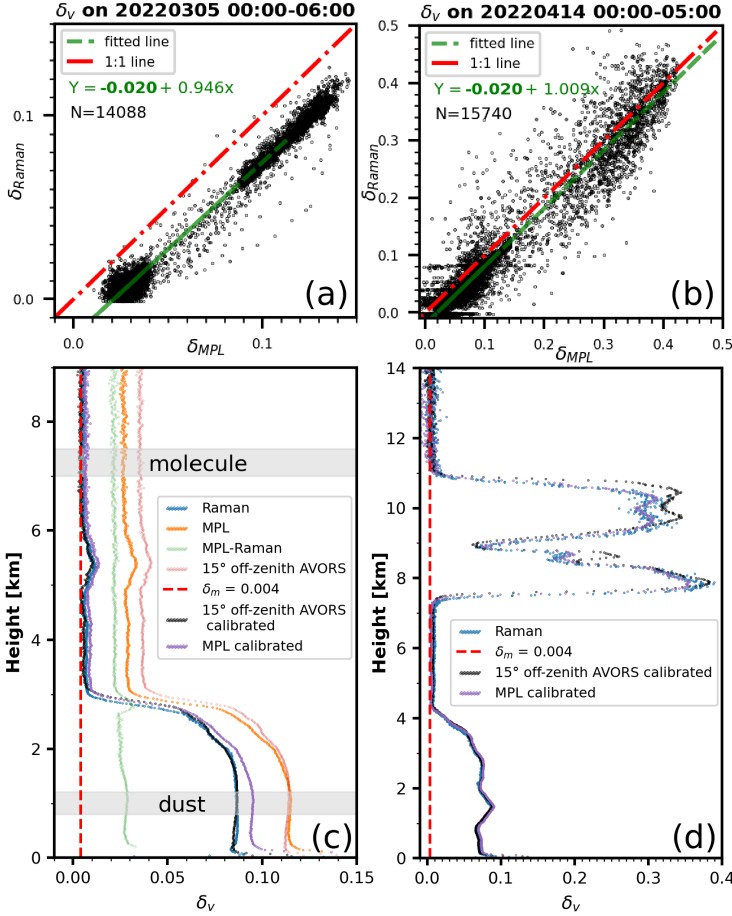

**Figure B1.** Scatter plot of well-calibrated Raman lidar and MPL uncalibrated depolarization ratio on (a) 5 March 2022, 00:00-06:00, Local Time and (b) 14 April 2022, 00:00-05:00, Local Time; (c) Averaged Depolarization profiles on 5 March 2022, 0:00-06:00; (d) Averaged Depolarization profiles on 14 April 2022, 00:00-05:00; The shade regions indicate the reference ranges used for dust and molecular layers.



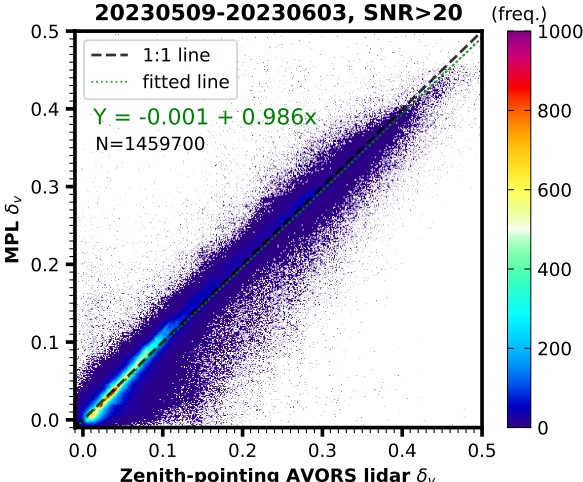

**Figure B2.** The density scatter plot between the zenith-pointing AVORS lidar and the MPL-calibrated depolarization ratio from 9 May 2023 to 3 June 2023, when two lidars are both zenith-pointing. Points including clouds, aerosols, and molecules with signal-to-noise ratios greater than 20 are used to plot the figure.

There is a period (from 9 May 2023 to 3 June 2023) when both MPL and AVORS lidar were zenith-pointing, we can utilize this period to validate our depolarization calibration result. Figure B2 shows the density scatter plot of MPL and AVORS lidar calibrated depolarization ratio of the cloud, aerosol, and molecule pixels observed during both lidar's zenith-pointing period.

The least-square squares fitting line (green dashed line) nearly overlaps with the 1:1 line (black dashed line), indicating well agreement of two lidars depolarization after calibration. According to the methods described above, we use the calibrated depolarization of MPL an AVORS lidar for the categorization algorithm and other analysis in this study.

**Appendix C: Description of RH calculation**

The vapor pressure $E$ could be calculated from the following equation:

$$E = \frac{rP}{0.622 + r} \tag{C1}$$

Where $r$ is the water vapor mixing ratio. Here we use the specific humidity ($q$) from ERA 5 data ($r = \frac{q}{1-q}$, since $q \ll 1$, $r \approx q$).

The saturated water vapor pressure with respect to liquid water $E_w$ and ice $E_i$ can be calculated from the following empirical formulas (Murray, 1967; Bolton, 1980):

$$E_w = 6.112 \exp\left[\frac{17.67(T - 273.16)}{T - 29.65}\right] \tag{C2}$$






$$E_{\mathrm{i}} = 6.1078\mathrm{exp}\left[\frac{21.87(T - 273.16)}{T - 7.66}\right] \tag{C3}$$

where $T$ is the temperature provided by radiosonde or ERA5. Then, relative humidity over water ($\mathrm{RH_w}$) and ice ($\mathrm{RH_i}$) could be calculated by

$$\mathrm{RH_w} = \frac{E}{E_{\mathrm{w}}} \tag{C4}$$


$$\mathrm{RH_i} = \frac{E}{E_{\mathrm{i}}} = \frac{\mathrm{RH_w}E_{\mathrm{w}}}{E_{\mathrm{i}}} \tag{C5}$$

For radiosonde, $\mathrm{RH_w}$ is provided, we use Eqs. (C2), (C3), and (C5) to get the $\mathrm{RH_i}$ profile. For ERA5 $\mathrm{RH_i}$ data, however, due to the ERA5 relative humidity is a piecewise function of saturation over water and ice (> 0°C over water, < -23°C over ice, between -23°C and 0°C interpolating the value over ice and water). We use the specific humidity to calculate the $\mathrm{RH_i}$ and $\mathrm{RH_w}$

separately (Eqs. (C1) to (C5)).

**Appendix D: Reynolds number and diameter calculation for falling ice crystals**

The diameter and Reynolds number of horizontally oriented ice crystals can be estimated from the terminal falling velocity measured by Doppler cloud radar for further discussion (He et al., 2021a). Here we use the approach proposed by Heymsfield and Westbrook (2010). First, as the columns have a negligible impact on specular reflection compared to the plates (Zhou et al.,

2012a), which present significantly larger surfaces to the incident lidar beam, we assume the HOIC shape is hexagonal plate and the thickness $h = 0.04D$, where $D$ is the diameter of ice crystal (Beard, 1980). The mass of a single ice crystal is then $m = \frac{3\sqrt{3}}{8}D^2 \times h \times \rho_{ice}$. The area ratio of ice crystal $A_r$ is define as $A_r = A/[(\pi/4)D^2]$. For horizontally oriented hexagonal plate ice crystal, $A_r = 0.827$.

The corresponding modified Davies number is defined as:

$$X = \frac{\rho_{air}}{\eta^2}\frac{8mg}{\pi A_r^{1-k}} \tag{D1}$$

where the optimum value of $k$ is 0.5. The density of air $\rho_{air}$ is the function of pressure ($P$) and temperature ($T$), and can be calculated by:

$$\rho_{air} = 1.293 \times \frac{P}{P_0} \times \frac{273.15}{T} \tag{D2}$$

where $P_0 = 1013.25$ hPa is the standard atmospheric pressure, $P$ and $T$ are interpolated from ERA5 data.

Air dynamic viscosity $\eta$ is defined as:

$$\eta = \eta_0 \times (\frac{T}{288.15})^{1.5} \times \frac{288.15 + B}{T + B} \tag{D3}$$





where $\eta_0$ is the air dynamic viscosity at 15 °C, $\eta_0 = 1.7894 \times 10^{-5}$ Pa· s (T = 15°C) and B is a gas-type-related constant, $B = 110.4$ K.

The Reynolds number $Re$ can be expressed as a function of Davies number:

$$Re = \frac{\delta_0^2}{4}\left[\left(1 + \frac{4\sqrt{X}}{\delta_0^2 \sqrt{C_0}}\right)^{1/2} - 1\right] \tag{D4}$$

where the inviscid coefficient $C_0 = 0.35$ and the dimensionless coefficient $\delta_0 = 8.0$. Finally, the crystal diameter D can be calculated through the following equation:

$$D = \frac{\eta Re}{\rho_{air} v_t} \tag{D5}$$

where $v_t$ is estimated by cloud radar Doppler velocity.

Finally, from Eqs. (D1), (D4), and (D5), we can retrieve the diameter of the horizontally oriented ice crystal and the corresponding Reynolds number. This method acts as an estimation to compare the case with the former researcher's study (Westbrook et al., 2010; He et al., 2021a).

## Appendix E: Calculation of eddy dissipation rates

The turbulence eddy dissipation rate (or turbulent kinetic energy dissipation rate, EDR, $\epsilon$) is a measure of the rate at which turbulent kinetic energy is converted into thermal energy due to viscous dissipation in a fluid. It quantifies how energy from larger turbulent eddies is transferred to smaller eddies and ultimately dissipated as heat, indicating the intensity of turbulence. The turbulence eddy dissipation rate was computed to reflect the turbulence using quantities including the standard deviation of Doppler velocity and horizontal wind speed in this study (Bouniol et al., 2003; O'Connor et al., 2010).

The standard deviation of the average wind serves as a measure of the kinetic energy present in turbulent scales that are typically larger than the size of the sampled volume. $\epsilon$ can be inferred from the variability in the vertical velocity over the 300 s sample time, $\sigma_{\overline{v}}$ in Eq.(E1) is the standard deviation of Doppler velocity within 300 s for 13 s unit time. By integrating the Kolmogorov (1941) turbulent energy spectrum formula within the inertial subrange, we can get:

$$\epsilon = \left(\frac{2}{3a}\right)^{3/2} \frac{\sigma_{\overline{v}}^3}{\left(k^{-2/3} - k_1^{-2/3}\right)^{3/2}} \tag{E1}$$

where $a = 0.55$ is one dimension Kolmogorov constant (Borque et al., 2016). And the wave number related to the large eddies traveling through the sampling volume during the sampling time is:

$$k = \frac{2\pi}{x_b + T_s|V_h|} \tag{E2}$$

where the width of the radar beam at height $z$ is $x_b = 2z\sin(\theta/2)$ with $\theta = 0.35°$ for this cloud radar. $|V_h|$ is the modulus of the horizontal wind, interpolated from ERA5 data. $T_s$ is the sampling time, 300 s in this case. And the wave number corresponding



to the length scale describing the scattering volume dimension for the dwell time for a single sample is given by:

$$k_1 = \frac{2\pi}{x_b + t|V_h|}$$ (E3)

Where dwell time $t = 13$ s for this cloud radar's case. In this way, the turbulence eddy dissipation rate was estimated from Eqs. (E1), (E2), and (E3). Note Bouniol et al. (2003) uses the time period of 30 s for $\sigma_{\bar{v}}$ compute and shows the estimated EDR are not sensitive to the number of points used. And a recent study (Nijhuis et al., 2019) points out that 10 min is still within the inertial sub-range, so we use the 300 s (5 min) time interval to calculate standard deviation considering the consistency with
other quantities' time resolution in this study.





## Appendix F: List of abbreviations and symbols

**Table F1.** List of abbreviations (acronyms), symbols and their explanations

| | |
|---|---|
| CALIPSO | Cloud-Aerosol Lidar and Infrared Pathfinder Satellite Observations |
| CALIOP | Cloud-Aerosol Lidar with Orthogonal Polarization |
| DSCOVR | Deep Space Climate Observatory |
| EDR | eddy dissipation rate |
| ECMWF | European Centre for Medium-Range Weather Forecast |
| EPIC | Earth Polychromatic Imaging Camera |
| ERA5 | ECMWF Reanalysis v5 |
| EarthCARE | Earth Cloud Aerosol and Radiation Explorer |
| HOIC | horizontally oriented ice crystal |
| LDR | linear depolarization ratio |
| MMCR | Millimeter-wave cloud radar |
| MPL | micro pulse lidar |
| MPC | mixed-phased cloud |
| NRB | normalized relative backscatter |
| PKU | Peking University |
| PMT | photomultiplier tube |
| POLDER | Polarization and Directionality of the Earth Reflectance |
| RH | relative humidity |
| ROIC | randomly oriented ice crystal |
| SNR | signal-to-noise ratio |
| SWC | supercooled water cloud |
| TKE | turbulent kinetic energy |
| $\beta'$ | attenuated backscatter |
| $\beta'_{\text{zenith}}$ | attenuated backscatter of zenith-pointing lidar |
| $\beta'_{\text{off-zenith}}$ | attenuated backscatter of off-zenith-pointing lidar |
| $\delta_v(\delta)$ | volume depolarization ratio |
| $\delta_{\text{zenith}}$ | volume depolarization ratio of zenith-pointing lidar |
| $\delta_{\text{off-zenith}}$ | volume depolarization ratio of off-zenith-pointing lidar |



*Data availability.* Radiosonde data can be obtained at http://weather.uwyo.edu/upperair/sounding.html (Beijing Radiosonde, 2022). The ERA5 data is available at https://cds.climate.copernicus.eu/. Lidar and radar data used to generate the results of this paper are available from the authors upon request (e-mail: ccli@pku.edu.cn).

*Author contributions.* ZW and CL conceived the research. ZW conducted the experiment, characterized the systems and analyzed the data. PS, AA, HB, CJ, and YH contributed to the scientific discussion. ZW, CJ and YH contributed to the development of the classification scheme. ZW wrote the manuscript guided by PS and HB. HB and CJ provided support with the depolarization calibration. ZW and HL took care of the radar-based products. YH helped with the microphysical retrieval. CL acquired the research funding. All co-authors contributed to proofreading of the manuscript.

*Competing interests.* The contact author has declared that none of the authors has any competing interest.

*Disclaimer.* Copernicus Publications remains neutral with regard to jurisdictional claims in published maps and institutional affiliations.

*Acknowledgements.* The authors thank the insightful discussions from Prof. Alexander Konoshonkin, Prof. Masanori Saito, and Prof. Andreas Macke. The authors thank AVORS Technology for providing lidar data, ECMWF for providing ERA5 data, and the University of Wyoming for providing the Beijing radiosonde data. We also thank the colleagues who participated in the operation of the lidar radar system
at our site.

*Financial support.* This work was supported in part by the National Natural Science Foundation of China/Research Grants Council Joint Research Projects under Grant 42075133, 42161160329, N_HKUST609/21, 42030607, 42305087, 42475095, Chinese Academy of Meteorological Sciences Basic Research Fund (Grants 2022Y008, 2023Z008), S&T Development Fund of Chinese Academy of Meteorological Sciences (Grant 2023KJ047) and in part by the National Natural Science Foundation of China under Grant 202306010350, the Chinese
Scholarship Council (CSC) and Alexander von Humbolt Foundation.



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
