# Peer review of "Assessment of horizontally-oriented ice crystals with a combination of multiangle polarization lidar and cloud Doppler radar"

_EGUsphere, 2024_

## Referee Comment (RC1)

Manuscript number: egusphere-2024-3841
Full title: Assessment of horizontally-oriented ice crystals with a combination of multiangle polarization lidar and cloud Doppler radar
Author(s): Wu Z. et al.

The authors proposed and performed a novel retrieval process to infer the horizontally oriented ice crystals (HOIC) using ground-based Doppler radar, zenith-pointing polarimetric lidar, and 15° off-zenith pointing polarimetric lidar. A combination of zenith-pointing and off-zenith-pointing lidars can provide range-resolved detections of HOICs in ice or mixed-phase clouds. The case study demonstrates a distinct relation between the abundance of HOICs and eddy dissipation rates inferred from collocated Doppler radar. In addition, correlations between HOICs and various environmental variables are explored. The present paper shows novel results regarding HOICs and the relationships between HOICs and dynamic and environmental variables and is suitable for *Atmospheric Measurement Techniques (AMT)*. However, the manuscript includes several insufficient descriptions and a lack of validation of some of the retrieval algorithms used in the present study. This manuscript requires major revisions before reconsideration of publication. Please find the comments below for potential improvements to the manuscript.

**Major comment**
1. Page 1, Line 3 in abstract and elsewhere "pixel": The terminology "pixel" is often used for a unit of the smallest area in the two-dimensional image data. For example, a satellite pixel indicates the smallest spatial area resolved by spaceborne spectrometers/imagers. It is a bit odd to use the terminology of "pixel" for a measured layer by active-sensor measurements, which is often referred to as a "range". To avoid any unnecessary confusion, I suggest the authors rephrase "pixel" with "range" throughout the text. In addition, a range-resolved algorithm for HOIC detection is not novel but was achieved by many previous studies (e.g., Noel and Sassen, 2005; Stilwell et al., 2019). It may be the first results based on a combination of zenith-pointing and 15° off-zenith pointing lidars, but it would be too specific to claim the first results. I suggest the authors simply remove the statement "for the first time".
2. On page 7, Lines 195-199, the authors discuss the horizontal deviation of the off-zenith pointing lidar. The discussions tacitly assume that the wind direction is along the line between the scattering volumes of the zenith-pointing and off-zenith-pointing lidars. This is not often the case in reality. The zenith-pointing lidar and off-zenith-pointing lidar often consistently measure a different portion of ice clouds, and therefore the time average does not justify these lidars observing the same portion of clouds. The authors must assume that clouds are horizontally homogeneous over a certain lateral scale, which is a strong assumption. Please clearly state the tacit assumption and discuss the validity of the assumption.
3. On page 18, the authors discuss the Euclidean distance from supercooled water clouds to ROICs and HOICs. The motivation behind this analysis is a bit questionable. First of all, the authors should separate the horizontal distances and vertical distances in the analyses as many microphysical processes (e.g., gravitational settling, ice aggregations, etc.) are reflected in the vertical distributions of cloud microphysical properties, and those of horizontal distributions may be influenced by a limited number of physical processes (e.g., turbulence, wind shear, etc.). With this in mind, the present analysis compares the distributions of Euclidean distances between HOICs and ROICs, which will unlikely to provide a meaningful interpretation as the distances in the discussion are an order of 10 km in contrast to the scales of physical processes

and scale of turbulence to be generally less than a few km. I suggest the authors remove the entire discussion regarding Euclidean distance.

4. Appendix D: The present analyses use the retrieval of ice crystal diameters as described in Appendix D. However, there are no descriptions of the uncertainty and potential bias in the estimated ice crystal diameters. The algorithm relies on a substantially simplified treatment of ice crystal shapes and orientations and is laid upon several approximations (e.g., aspect ratios). The authors should discuss the accuracy of the retrieval method in Appendix D.

**Minor comments**

1. Page 2, Lines 31-32 "Mie scattering …": Mie scattering theory applies to spheres and cannot examine the differences in the scattering cross-sections between random orientation and preferential orientations (i.e., particle orientations cannot be defined). Please clarify the point of the statement.
2. Page 3, Line 60: "didn't" should be "did not".
3. Page 3, Lines 71-72: "Westbrook et al. 2010" Suggest the authors add "Sato and Okamoto (2011)."
4. Page 3, Line 72: "Zhou et al., 2012a" Suggest the authors add "Saito et al., (2017)."
5. Page 5, Line 123 "cos(75°)": It would be better to use a unit of steradian inside the cosine. By the way, should this be 15°? Cos(15°) is a very small value.
6. Page 13, Lines 307-309 "… the strong turbulence caused by the latent heat released due to the sublimation…": This statement lacks supporting evidence and is not beyond the speculation level. Please provide sufficient evidence supporting this or clearly state that this is based on the authors' speculation.
7. Page 15, Line 334 "A negative correlation is found …": Is there a hypothesized mechanism for the negative correlation? Also, is this true for ROICs or not?
8. Figure 4 and Page 18, Line 370: I am concerned with the consistency of the scattering volumes between radar and lidars. Between 16:00 and 18:00 in Fig. 4, the liquid layer appears at an altitude of 5-6 km, as evidenced by the strong echoes from both lidars. However, it is not seen from radar. Please discuss the scattering volume consistencies. Perhaps the authors need to discuss the minimum detectable radar reflectivity in Section 2.4.
9. Figure 4 and Page 18, Lines 375–376 "… the signals of both lidar systems were subject to strong attenuation": It is hard to see the attenuation from Fig. 4 which uses a linear scale in the attenuated backscattering coefficients. Suggest the authors use a log scale in this figure.

**Reference**

Noel, V., & Sassen, K. (2005). Study of planar ice crystal orientations in ice clouds from scanning polarization lidar observations. *Journal of Applied Meteorology*, *44*(5), 653-664.

Saito, M., Iwabuchi, H., Yang, P., Tang, G., King, M. D., & Sekiguchi, M. (2017). Ice particle morphology and microphysical properties of cirrus clouds inferred from combined CALIOP-IIR measurements. *Journal of Geophysical Research: Atmospheres*, *122*(8), 4440-4462.

Sato, K., & Okamoto, H. (2011). Refinement of global ice microphysics using spaceborne active sensors. *Journal of Geophysical Research: Atmospheres*, *116*(D20).

Stillwell, R. A., Neely III, R. R., Thayer, J. P., Walden, V. P., Shupe, M. D., & Miller, N. B. (2019). Radiative influence of horizontally oriented ice crystals over summit, Greenland. *Journal of Geophysical Research: Atmospheres*, *124*(22), 12141-12156.

---

## Author Comment (AC1)

**Responses to Referee #1's comments**

Manuscript number: egusphere-2024-3841
Full title: Assessment of horizontally-oriented ice crystals with a combination of multiangle polarization lidar and cloud Doppler radar Author(s): Wu Z. et al.

The authors proposed and performed a novel retrieval process to infer the horizontally oriented ice crystals (HOIC) using ground-based Doppler radar, zenith-pointing polarimetric lidar, and 15° off zenith-pointing polarimetric lidar. A combination of zenith-pointing and off-zenith-pointing lidars can provide range-resolved detections of HOICs in ice or mixed-phase clouds. The case study demonstrates a distinct relation between the abundance of HOICs and eddy dissipation rates inferred from collocated Doppler radar. In addition, correlations between HOICs and various environmental variables are explored. The present paper shows novel results regarding HOICs and the relationships between HOICs and dynamic and environmental variables and is suitable for *Atmospheric Measurement Techniques (AMT)*. However, the manuscript includes several insufficient descriptions and a lack of validation of some of the retrieval algorithms used in the present study. This manuscript requires major revisions before reconsideration of publication. Please find the comments below for potential improvements to the manuscript.

**Reply:** We sincerely thank Referee #1 for reviewing our paper and providing constructive comments for improvement. Responses to these comments are provided below. In this author's comment, we reply (in blue font) to Referee #1's comments (in black font). The relevant part in the revised version is in green font.

**Major comment**

1. Page 1, Line 3 in abstract and elsewhere "pixel": The terminology "pixel" is often used for a unit of the smallest area in the two-dimensional image data. For example, a satellite pixel indicates the smallest spatial area resolved by spaceborne spectrometers/imagers. It is a bit odd to use the terminology of "pixel" for a measured layer by active-sensor measurements, which is often referred to as a "range". To avoid any unnecessary confusion, I suggest the authors rephrase "pixel" with "range" throughout the text. In addition, a range-resolved algorithm for HOIC detection is not novel but was achieved by many previous studies (e.g., Noel and Sassen, 2005; Stilwell et al., 2019). It may be the first results based on a combination of zenith-pointing and 15° off-zenith pointing lidars, but it would be too specific to claim the first results. I suggest the authors simply remove the statement "for the first time".

**Reply:** Thank you for the critical comment!
The terminology "pixel" can be seen in other lidar-based literatures like:
Cloudnet related: Hogan and Ewan (2004), Schimmel et al. (2022)
PollyNET: Baars et al. (2017)
EarthCARE: Zadelhoff et al. (2023). Donovan et al. (2024)

➢  To avoid any unnecessary confusion, we have changed the terminology "pixel" to "range", "range bin" or "bin" in different positions. And we have changed "pixel-by-pixel" to "range-resolved".

➢  We have removed the statement "for the first time" both in the abstract and Line 91.

2.  On page 7, Lines 195-199, the authors discuss the horizontal deviation of the off-zenith pointing lidar. The discussions tacitly assume that the wind direction is along the line between the scattering volumes of the zenith-pointing and off-zenith-pointing lidars. This is not often the case in reality. The zenith-pointing lidar and off-zenith-pointing lidar often consistently measure a different portion of ice clouds, and therefore the time average does not justify these lidars observing the same portion of clouds. The authors must assume that clouds are horizontally homogeneous over a certain lateral scale, which is a strong assumption. Please clearly state the tacit assumption and discuss the validity of the assumption.

**Reply:** Thank you for pointing out this. The wind direction is not always along the line between the scattering volumes of the two-angle lidars.
    The off-zenith-pointing lidar was pointing towards the due north to avoid the noise of sunlight to the greatest extent. The wind rose plot (wind direction distribution) in Beijing radiosonde station in the whole year of 2022 is shown in Fig. S1. Two kinds of criteria are used to select the height and temperature where HOICs tend to exist: (a) altitude between 4 km and 8 km; (b) temperature between -30 ℃ and -5 ℃. From the wind rose frequency diagram, we found the dominant wind direction over Beijing where HOICs are found is west and northwest, which can also be seen in Fig. 6b.

[Figure]

Figure S1. The wind rose plot of Beijing radiosonde station, the whole year 2022. (a) The altitude between 4 km and 8 km was selected. (b) The temperature between -30 ℃ and -5 ℃ was selected. Around 20, 000 radiosonde data points are used to create this plot.

In this way, the wind direction was only sometimes (when the wind vector has a north or south component) along the line between the scattering volumes of the two-angle lidars.

In the revised text, we have clearly added (Lines 197 -205 in the revised manuscript) the assumption of horizontal homogeneity of the clouds, as shown below. This is a strong assumption, and all research based on a two-angle lidar scheme faces this problem (Westbrook et al., 2010; He et al., 2021).

"Assuming a horizontal wind speed is $v = 20$ m s$^{-1}$ (see radiosonde Fig. 6b) and the wind direction is along the line between the scattering volumes of the two angle lidars, the horizontal movement of the cloud is 6000 m within five minutes, which is the temporal resolution utilized in data processing. Consequently, if both lidars observe the same cloud within the same time slot (> 5 min), the horizontal deviation of the off-zenith pointing lidar is less significant (1.6 km < 6 km). Although with increasing height, the horizontal distance between the probed volumes also increases (from 0.268 km at 1 km height to 2.68 km at 10 km height.). In reality, the wind direction does not always align with the line connecting the scattering volumes of the two-angle lidars. Therefore, we must assume horizontal homogeneity of the detected cloud layers over a certain lateral scale. This assumption is likely valid for horizontally homogeneous stratiform clouds. However, caution is needed for discrete, small-scale clouds, as misalignment may occur."

3. On page 18, the authors discuss the Euclidean distance from supercooled water clouds to ROICs and HOICs. The motivation behind this analysis is a bit questionable. First of all, the authors should separate the horizontal distances and vertical distances in the analyses as many microphysical processes (e.g., gravitational settling, ice aggregations, etc.) are reflected in the vertical distributions of cloud microphysical properties, and those of horizontal distributions may be influenced by a limited number of physical processes (e.g., turbulence, wind shear, etc.). With this in mind, the present analysis compares the distributions of Euclidean distances between HOICs and ROICs, which will unlikely to provide a meaningful interpretation as the distances in the discussion are an order of 10 km in contrast to the scales of physical processes and scale of turbulence to be generally less than a few km. I suggest the authors remove the entire discussion regarding Euclidean distance.

**Reply:** Thank you for pointing out this.
We acknowledge the concerns raised by the Reviewer. Our approach was motivated by the intrinsic need to provide a quantitative measure of the linkage between the presence of liquid water and the occurrence of HOIC. So far, earlier studies only qualitatively elaborated about this relationship (e.g., Westbrook et al., 2010). A quantification of an impact of supercooled liquid water on HOIC generation will be of high value for the community as it would provide an important constraint for HOIC-generating conditions. To our opinion, the presented approach to evaluate the Euclidean distance between certain cloud features (in our case the occurrence of HOIC or ROIC) is thus worth to be introduced. In order to motivate the approach better, we introduced further text to the beginning of Section 4.5 (lines 402-411 of the revised manuscript):

"While this attempt is promising based on case studies of well-defined scenarios, such as for ice formation in stratiform supercooled liquid clouds, a statistically comprehensive approach that covers the full variety of cloud types is challenging. One reason is that often the lidar signal is attenuated already within the ice virgae below, so that no signatures of liquid-dominated ice forming layers can be observed. Cloud radar techniques, in turn, are frequently not sensitive enough to detect layers of liquid water. Second reason is, that the ice-forming supercooled liquid water layers might eventually disappear due to cloud dynamical or microphysical processes, while the formed ice particles still exist. A third reason is that vertical wind shear and the microphysical evolution of the ice particles during falling blur the signatures of potential direct relationships between liquid layers and HOIC occurrence.
In here, we introduce the application of the Euclidean distance between supercooled liquid water bins and HOIC or ROIC, respectively, as an approach to quantify the impact of supercooled liquid water on HOIC formation."

In order to consider the concerns raised by the Reviewer, we introduced another short statement to the conclusions section, where it is now emphasized that this approach is only a starting point for more comprehensive future investigations (lines 454-456 of the revised manuscript)

"We see a high potential in using the Euclidean distance approach, even though an improved quantification will require an enhanced characterization of the presence of liquid water beyond lidar attenuation (e.g., Schimmel et al., 2022) and an improved consideration of the ice crystal evolution during sedimentation (Vogl et al., 2024)."

4. Appendix D: The present analyses use the retrieval of ice crystal diameters as described in Appendix D. However, there are no descriptions of the uncertainty and potential bias in the estimated ice crystal diameters. The algorithm relies on a substantially simplified treatment of ice crystal shapes and orientations and is laid upon several approximations (e.g., aspect ratios). The authors should discuss the accuracy of the retrieval method in Appendix D.

**Reply:** Thank you for pointing out this. We try to analyze the uncertainty of retrieval. This method was widely used in previous studies (Westbrook et al. 2010; He et al. 2021). Three main points are considered as the main primary sources of uncertainty.

1. As you suggested, first we try to change the aspect ratio. Currently, we use the fixed aspect ratio ($h/D$, height divided by diameter) of 0.04 for the hexagonal plate. Now, we do the sensitivity tests for the assumed aspect ratio. Change the aspect ratios for the hexagonal plate from 0.01 to 0.2 (Beard 1980; Stout et al., 2024).
   First, we consider a fixed falling velocity of 0.8 ms$^{-1}$(median Doppler velocity found in this study). Fix the temperature and air pressure condition as of 13 October 2022. The corresponding mean Re and $D$ are shown in Table 1, the retrieved Re and $D$ are rather sensitive to the assumed fixed aspect ratio.

Table 1. Estimated mean Reynolds number and crystal diameter corresponding to a fall velocity of 0.8 ms$^{-1}$ using a fixed aspect ratio.

| Aspect ratio | Re | D [μm] |
|---|---|---|
| 0.01 | 102 | 3041 |
| **0.04** | **40** | **1207** |
| 0.08 | 26 | 779 |
| 0.1 | 23 | 679 |
| 0.2 | 15 | 446 |

To minimize the possible uncertainty introduced by the fixed aspect ratio, instead, we use the dynamic empirical relationships (functions of diameter) used by the previous studies:

Saito et al. (2019)

$$\frac{h}{D} = \left( 0.8038 \left( \frac{D}{2} \right)^{0.526} \right)^{-1}, D > 10 \text{ (μm)} \tag{1}$$

Bréon et al. (2004)

$$\frac{h}{D} = 2.01 D^{-0.551} \text{ (μm)} \tag{2}$$

From Eqs. (D1), (D4) and (D5), we combine Eqs (1) and (2), respectively. The estimated Reynolds number and crystal diameters are calculated and shown in Table 2.

Table 2. Estimated mean Reynolds number and crystal diameter corresponding to a fall velocity of 0.8 ms$^{-1}$ using different empirical aspect ratio relationships.

| Empirical h/D relationship | Calculated Aspect ratio | Re | D [μm] |
|---|---|---|---|
| Saito et al., 2019 | 0.044 | 38 | 1129 |
| Bréon et al., 2004 | 0.043 | 40 | 1199 |

The corresponding retrieved Re and D are quite close to the results using the fixed aspect ratio of 0.04, which confirms the validity of using a fixed aspect ratio of 0.04.

Next, instead of using a fixed median falling velocity, we use the real detected Doppler velocity (a distribution of velocity with a median value of 0.8 ms$^{-1}$), the corresponding distribution and statistics of Re and D are shown below in Fig. S2 and Table 3:

[Figure]

Figure S2. (a) The histogram of retrieved diameters of HOICs. (b) The histogram of retrieved Reynolds numbers of HOICs. Dynamic empirical relationships and fixed different aspect ratios (from 0.01 to 0.2) are used for $D$ and $Re$ retrievals.

Table 3. Different aspect ratios: statistics of estimated diameter and Reynolds number for HOICs on 13 October 2022.

| Aspect ratio | Statistics | Re | $D$ [µm] |
|---|---|---|---|
| 0.04 | 5th percentile | 28 | 1029 |
| | First quartile | 39 | 1204 |
| | Median | 51 | 1354 |
| | Third quartile | 65 | 1525 |
| | 95th percentile | 88 | 1756 |
| | Mean | 54 | 1369 |
| Saito et al., 2019 | 5th percentile | 24 | 894 |
| | First quartile | 37 | 1125 |
| | Median | 50 | 1342 |
| | Third quartile | 69 | 1611 |
| | 95th percentile | 101 | 2016 |
| | Mean | 55 | 1387 |
| Bréon et al., 2004 | 5th percentile | 26 | 941 |
| | First quartile | 39 | 1194 |
| | Median | 54 | 1434 |
| | Third quartile | 74 | 1735 |
| | 95th percentile | 110 | 2193 |
| | Mean | 59 | 1487 |

From Fig. S2 and Table 3, we can conclude that using a fixed aspect ratio of 0.04 leads to limited uncertainty compared to the empirical aspect ratio relationships used in literature (Bréon et al., 2004; Saito et al., 2019). The median and mean retrieved diameters and Reynolds numbers obtained with the fixed aspect ratio of 0.04 are particularly close to those derived from Saito et al. (2019)'s empirical dynamic relationships. The only notable difference is that the distributions of Re and $D$ from the dynamic relationship are wider.

2.  The shape of the particle also contributes to the uncertainty of the retrieval. The hexagonal plate is the most simplified shape, widely used for oriented ice (Bréon et al., 2004; Zhou et al., 2012; He et al., 2021).  As Westbrook et al. (2010) have already shown in their Table III. The crystal diameter corresponding to a certain falling velocity is rather sensitive to the assumed shape of ice crystals. To make it concise and comparable to the previous research, we kept using hexagonal plates.

3.  The assumption that Doppler velocity represents the terminal velocity of a particle is a rough approximation. However, in general, long-term measured Doppler velocity can partially mitigate the effect of rapidly changing vertical airflow and provide an approximate still-air velocity for falling ice crystals. A reasonable approach to reducing uncertainty is to remove extreme values from the retrieved Reynolds numbers and diameters. Therefore, we focus on the intermediate range of the retrieved diameters and Reynolds numbers, excluding extreme values (e.g., data points beyond the 5th and 95th percentiles).

We have added a discussion part about the uncertainty in Appendix D in terms of the above three points:

"After conducting careful sensitivity tests, we found that the assumed fixed aspect ratio of 0.04 yields a retrieved diameter and Reynolds number similar to those obtained using empirical dynamic aspect ratio relationships reported in the literature (Bréon et al., 2004; Saito et al., 2019). It is important to note that crystal diameter and Reynolds number are highly sensitive to the shape of ice crystals (Westbrook et al., 2010, Table III ). The assumed HOIC shape of hexagonal plates is the most simplified and widely used model (Bréon et al., 2004; Zhou et al., 2012; He et al., 2021). Additionally, assuming that Doppler velocity represents the terminal velocity of a particle in still air introduces some uncertainty. However, in general, long-term Doppler velocity measurements can partially mitigate the effect of rapidly changing vertical airflow and provide an approximate still-air velocity for falling ice crystals. To reduce the potential uncertainty caused by extreme vertical airflow, we focus only on the intermediate range of the retrieved diameters and Reynolds numbers, excluding extreme values (e.g., data points beyond the 5th and 95th percentiles). In summary, this method serves as an estimation to compare the case with previous studies (Westbrook et al., 2010; He et al., 2021a)."

**Minor comments**

1. Page 2, Lines 31-32 "Mie scattering …": Mie scattering theory applies to spheres and cannot examine the differences in the scattering cross-sections between random orientation and preferential orientations (i.e., particle orientations cannot be defined). Please clarify the point of the statement.

   **Reply:** Thank you for pointing out this! This viewpoint is originally from the penultimate paragraph of Várnai et al. (2019).
   We have rephased the sentence in terms of this by removing the "Mie scattering":

   "Calculation shows oriented plates intercept roughly twice as much sunlight as the perfectly randomly oriented ones (Várnai et al., 2019)."

2. Page 3, Line 60: "didn't" should be "did not".

   **Reply:** Done.

3. Page 3, Lines 71-72: "Westbrook et al. 2010" Suggest the authors add "Sato and Okamoto (2011)."

   **Reply:** Done. We have added Sato and Okamoto (2011).

4. Page 3, Line 72: "Zhou et al., 2012a" Suggest the authors add "Saito et al., (2017)."

   **Reply:** Done. We have added Saito et al., (2017).

5. Page 5, Line 123 "cos(75°)": It would be better to use a unit of steradian inside the cosine. By the way, should this be 15°? Cos(15°) is a very small value.

   **Reply:** Thank you for pointing out this. It should be 15° or $\frac{\pi}{12}$ in the unit of steradian. We used $\cos\left(\frac{\pi}{12}\right)$ in Line 123. And we also used $\tan\left(\frac{\pi}{12}\right)$ in Line 195.

6. Page 13, Lines 307-309 "… the strong turbulence caused by the latent heat released due to the sublimation…": This statement lacks supporting evidence and is not beyond the speculation level. Please provide sufficient evidence supporting this or clearly state that this is based on the authors' speculation.

   **Reply:** This is from the authors' speculation. We have clearly stated this in the revised version Line 314-316:

   "The high $\delta_v$ region above the cloud base (Fig. 5b, brown line within the red shaded region, also see Fig. 4d) in the zenith lidar observation appears to be associated with

a higher eddy dissipation rate (Fig. 5e, also see Fig. 4j), suggesting that the strong turbulence may be linked to latent heat release from the sublimation of ice crystals near the cloud base."

7.  Page 15, Line 334 "A negative correlation is found …": Is there a hypothesized mechanism for the negative correlation? Also, is this true for ROICs or not?

**Reply:** Overall, within the troposphere, the higher the altitude, the lower the temperature, and the higher the horizontal wind speed. The horizontal wind and the temperature are negatively correlated. The description here is to confirm the negative correlation relationship for the environment variable of HOICs. That is also true for ROICs, as shown in Fig. S3 below:

[Figure]

Figure S3. The density scatter plot of horizontal wind speed and temperature where ROICs exist, the greener the color, the higher the number density of ROIC pixels

8.  Figure 4 and Page 18, Line 370: I am concerned with the consistency of the scattering volumes between radar and lidars. Between 16:00 and 18:00 in Fig. 4, the liquid layer appears at an altitude of 5-6 km, as evidenced by the strong echoes from both lidars. However, it is not seen from radar. Please discuss the scattering volume consistencies. Perhaps the authors need to discuss the minimum detectable radar reflectivity in Section 2.4.

**Reply:** Yes, the minimum detectable radar reflectivity is about -40 dBZ. Some small supercooled liquid water droplets are too small, they can be detected by the lidars and are not detectable by radar (especially for the Ka-band radars compared with W-band radars which have a shorter wavelength and are more sensitive to smaller particles).
    We have added the discussion of the scattering volume consistencies on Page 18, Line 399-400:
    "It should be noted that the scattering volume of lidars and radar is not exactly the same. Small liquid droplets and optically thin ice clouds are sometimes not detectable from Ka-band radar compared with lidars."

`

We have added the minimum detectable radar reflectivity discussion in Section 2.4.

"The minimum detectable reflectivity factor of this radar is -40 dBZ. Compared to lidars, radar exhibits greater sensitivity to larger particles (Westbrook et al., 2010; Bian et al., 2023). However, this Ka-band cloud radar may fail to detect certain tiny liquid droplets and optically thin ice clouds."

9. Figure 4 and Page 18, Lines 375–376 "… the signals of both lidar systems were subject to strong attenuation": It is hard to see the attenuation from Fig. 4 which uses a linear scale in the attenuated backscattering coefficients. Suggest the authors use a log scale in this figure.

**Reply:** Thank you for pointing out this. We have changed Figs.4a and 4c (Fig. S4 below) to log scale to show the attenuation in the revised manuscript.

[Figure]

Figure S4. Lidar ((a)-(g)) and zenith-pointing Ka-band cloud radar ((h)-(l)) observations on 13 October 2022, time-height contour plots (5 min / 15 m resolution for (a)-(g), 13 s / 30 m resolution for (h)(i) to show the variation of Doppler velocity, 5 min / 30 m for (j)-(l)). (a) 15 ◦ off-zenith-pointing lidar attenuated backscatter. (b) 15 ◦ off-zenith-pointing lidar volume depolarization ratio. (c) Zenith-pointing lidar attenuated backscatter. (d) Zenith-pointing lidar volume depolarization ratio. (e) The ratio of attenuated backscatter for zenith-pointing and off-zenith-pointing lidar. (f) The ratio of volume depolarization ratio for zenith-pointing and off-zenith-pointing lidar. (g) Cloud phase categorization results with isotherm from ERA 5 data. Abbreviations of SWC, ROIC, HOIC, and MPC represent supercooled liquid water cloud, randomly oriented ice crystal, horizontally oriented ice crystal, and mixed-phased cloud. There is no cloud pixel categorized as (warm) water due to the subzero temperature. (h)(i)(k)(l) Cloud radar detected momentum data: Doppler velocity, spectral width, reflectivity (with isotherm from ERA 5 data), and linear depolarization ratio (LDR). (j) Cloud radar retrieved eddy dissipation rate (EDR, ε).

**Reference**

Baars, H., Seifert, P., Engelmann, R., and Wandinger, U.: Target categorization of aerosol and clouds by continuous multiwavelength-polarization lidar measurements, *Atmos. Meas. Tech., 10*, 3175–3201, https://doi.org/10.5194/amt-10-3175-2017, 2017.

Beard, K. V.: The Effects of Altitude and Electrical Force on the Terminal Velocity of Hydrometeors. *J. Atmos. Sci.*, **37**, 1363–1374, https://doi.org/10.1175/1520-0469(1980)037<1363:TEOAAE>2.0.CO;2, 1980.

Bian, Y., Liu, L., Zheng, J., Wu, S., & Dai, G: Classification of cloud phase using combined ground-based polarization LiDAR and millimeter cloud radar observations over the Tibetan Plateau. *IEEE Transactions on Geoscience and Remote Sensing*, *61*, 1-13, https://doi.org/10.1109/LGRS.2019.2930866, 2023

Bréon, F.-M. and Dubrulle, B.: Horizontally oriented plates in clouds, *Journal of the atmospheric sciences, 61*, 2888–2898, https://doi.org/10.1175/JAS-3309.1, 2004.

Donovan, D. P., van Zadelhoff, G.-J., and Wang, P.: The EarthCARE lidar cloud and aerosol profile processor (A-PRO): the A-AER, A-EBD, A-TC, and A-ICE products, Atmos. Meas. Tech., 17, 5301–5340, https://doi.org/10.5194/amt-17-5301-2024, 2024.

He, Y., Liu, F., Yin, Z., Zhang, Y., Zhan, Y., and Yi, F.: Horizontally oriented ice crystals observed by the synergy of zenith-and slant-pointed polarization lidar over Wuhan (30.5° N, 114.4° E), China, *Journal of Quantitative Spectroscopy and Radiative Transfer, 268,* 107 626, 675 https://doi.org/10.1016/j.jqsrt.2021.107626, 2021.

Hogan, Robin J., and Ewan J. O'Connor. Facilitating cloud radar and lidar algorithms: the Cloudnet Instrument Synergy/Target Categorization product. *Cloudnet documentation* 14, 2004.

Saito, M. and Yang, P.: Oriented ice crystals: a single-scattering property database for applications to lidar and optical phenomenon simulations, *Journal of the Atmospheric Sciences, 76*, 2635–2652, https://doi.org/10.1175/JAS-D-19-0031.1, 2019

Schimmel, W., Kalesse-Los, H., Maahn, M., Vogl, T., Foth, A., Garfias, P. S., and Seifert, P.: Identifying cloud droplets beyond lidar attenuation from vertically pointing cloud radar observations using artificial neural networks, *Atmos. Meas. Tech., 15*, 5343–5366, https://doi.org/10.5194/amt-15-5343-2022, 2022.

Stout, J. R., Westbrook, C. D., Stein, T. H. M., and McCorquodale, M. W.: Stable and unstable fall motions of plate-like ice crystal analogues, *Atmos. Chem. Phys., 24,* 11133–11155, https://doi.org/10.5194/acp-24-11133-2024, 2024.

van Zadelhoff, G.-J., Donovan, D. P., and Wang, P.: Detection of aerosol and cloud features for the EarthCARE atmospheric lidar (ATLID): the ATLID FeatureMask (A-FM) product, *Atmos. Meas. Tech., 16*, 3631–3651, https://doi.org/10.5194/amt-16-3631-2023, 2023.

Várnai, T., Kostinski, A. B., and Marshak, A.: Deep space observations of sun glints from marine ice clouds, *IEEE Geoscience and Remote Sensing Letters, 17*, 735–739, https://doi.org/10.1109/LGRS.2019.2930866, 2019

Westbrook, C., Illingworth, A., O'Connor, E., and Hogan, R.: Doppler lidar measurements of oriented planar ice crystals falling from supercooled and glaciated layer clouds, *Quarterly Journal of the Royal Meteorological Society, 136*, 260–276, https://doi.org/10.1002/qj.528, 2010.

Zhou, C., Yang, P., Dessler, A. E., and Liang, F.: Statistical properties of horizontally oriented plates in optically thick clouds from satellite observations, IEEE Geoscience and Remote Sensing Letters, 10, 986–990, https://doi.org/10.1175/JAMC-D-11-0265.1, 2012

---

## Author Comment (AC2)

**Responses to Referee #2's comments**

The manuscript presents an automatic algorithm for horizontal ice crystal (HOIC) identification using the combination of the attenuated backscatter and volume depolarization ratio of HOICs for zenith and 15°-off-zenith lidar measurements. Moreover, using Doppler radar measurements, the authors provide insights of the physical processes in clouds with HOICs.

In general, the work is well-presented and the results seem quite promising towards the identification of HOICs and the description of the physical processes. There are though several aspects that require more attention, as for example the points highlighted from Referee #1. In addition, a more appropriate characterization of the lidar measurements should be provided. Since the algorithm is heavily based on the volume depolarization ratio measurements from the zenith (MPL) and off-zenith (AVORS) lidars, more details need to be provided regarding the calibration procedure and the uncertainties of the volume depolarization measurements:

**Reply:** We sincerely thank Referee #2 for reviewing our paper and providing constructive comments for improvement. Responses to these comments are provided below. In this author's comment, we reply (in blue font) to Referee #1's comments (in black font and key points in red font). The relevant part in the revised version is in green font.

Revisions regarding lidar calibration:

1. Since the identification algorithm is heavily-based on the values of the volume depolarization ratio measured at zenith with MPL lidar and at 15°-off-zenith with AVORS lidar, the calibration of these systems should have been done following higher standards than the comparison with a reference system. For example, why the procedures according to established lidar networks (e.g. EARLINET) or the extensive work of Freudenthaler (2016) were not followed?

**Reply:**

In principle, we should use the best calibration method for the depolarization calibration. The accurate depolarization is the first step for an identification method using the depolarization ratio. We did the calibration at the beginning of our research.

However, the two lidars, MPL and AVORS, used here are compact and commercial, small-sized lidars. The height of both lidars is below one meter (see Fig. S4). For this reason, an adaptation of the system to perform Δ90 calibrations, as described by Freudenthaler (2016), has not been possible.

The reference Raman lidar is relatively large and is wrapped inside a container. For this system the Δ90 depolarization calibration procedure is conducted regularly to confirm the validity of the depolarization ratio. The depolarization performance of this Raman lidar has also been validated by another collocated high spectral resolution lidar (HSRL) (Wang et al., 2022). In general, it is a reliable lidar as a reference system.

For such compact systems, the calibration using a reference system has shown promising results (Córdoba-Jabonero et al. 2021; Papetta et al. 2024). Papetta et al. (2024) introduced an empirical method to perform this type of calibration, which allows to consider changes in the

inter-channel gain ratio factor and also polarization effects in the system. We followed this approach and were able to match the depolarization ratio of the three systems, i.e., the calibrated Raman system, and the two compact systems (MPL and AVORS). In the case of the MPL the same detector is used to measure the co and cross polarized components keeping the gain factor over time stable, and the comparison of the MPL and AVORS system in the molecular region in the long term corroborates the stability of the AVORS's gain ratio as well.

We added the description in the revised manuscript, Line 500-501:

"Since the MPL and AVORS lidar are both compact, small-sized lidars, the standard Δ90 method (Freudenthaler 2016) is not applicable."

2.    As shown in Fig. B1c and d, the "MPL-calibrated" and "AVORS-calibrated" volume depolarization ratios show differences with the reference Raman lidar, that indicate possible limitations in the calibration procedure used. In addition, Fig. B2 shows the differences for the volume depolarization ratio when both lidars measure at the zenith direction. The authors should justify the differences, providing an analysis for the cases associated with the larger ones shown in the plots (especially in Fig. B2). Moreover, in line 478 the authors refer to the differences shown in Fig. B1 as "acceptable": First, the authors need to provide the uncertainties of volume depolarization ratio profiles for all three lidars (MPL, AVORS, Raman), and include them in the plots. Then, they need to provide the larger differences that are "acceptable" for the algorithm presented, and discuss whether the differences shown in Fig. B1 and B2 are smaller than these thresholds.

**Reply:**

➢   For the Fig. B1c, the main difference is from the calibrated MPL and Raman lidar depolarization ratio below 2 km. The difference is about 1%. Within 2 km, there is always an effect of overlap calibration uncertainty which affects the depolarization ratio for Raman lidar and AVORS lidar (but not in MPL because it only has one optical path), and it usually can not be solved 100%. There could also be deadtime effects for MPL in the near range that might deviate a bit the depolarization.

➢   In Fig. B1d, the depolarization ratios measured by the MPL and Raman lidars are nearly identical, with a maximum difference of approximately 2%. However, the AVORS lidar, operating at a 15-degree off-zenith angle, exhibits a higher depolarization ratio within the ice cloud layer, likely due to the presence of horizontally oriented ice crystals, compounded by spatial heterogeneity. These oriented crystals may significantly enhance the depolarization signal in AVORS measurements compared to the zenith-pointing MPL. Additionally, the differing fields of view (FOV)—0.1 mrad for MPL and 0.2 mrad for AVORS—contribute to slightly distinct multiple scattering effects within dense clouds, further amplifying the observed differences.

➢   Provide analysis in Figure B2 difference

The results of the depolarization calibration, both before and after, are shown in Fig. S1 using density scatterplots.

[Figure]

**Figure S1.** Density scatterplots of depolarization ratio ($\delta_v$) before (a, c) and after (b, d) depolarization calibration for MPL and zenith-pointing AVORS lidar from 9 May 2023 to 3 June 2023, when two lidars were both zenith-pointing. Data include clouds, aerosols, and molecules with signal-to-noise ratios greater than 20 (a, b) or 30 (c, d). The 1:1 line and least-squares regression fits are shown for comparison.

From Fig. S1, it is evident that the consistency between the depolarization measurements of the two lidars improved significantly after calibration. The intercept of the least squares fit becomes closer to zero following the calibration. However, since the calibration is a linear transformation, it cannot remove the noisy points that deviate from the 1:1 line. As the AVORS lidar was positioned outside the container, it was more susceptible to solar noise, resulting in a lower signal-to-noise ratio (SNR). This may also contribute to the observed discrepancies. By increasing the SNR threshold from 20 to 30 for point selection, we observed a better agreement in the calibrated depolarization ratio (compare Fig. S1b and S1d).

A representative case was observed on 21 May 2023, featuring multiple ice cloud layers and a dust layer, which highlights the effectiveness of the calibrated lidar depolarization

measurements (Fig. S2). The two zenith-pointing lidars show excellent agreement across both high and low depolarization regions throughout the 0–14 km observation range.

[Figure]

Figure S2 The zenith-pointing MPL and zenith-pointing AVORS lidar volume depolarization ratio profiles on 21 May 2023, (a) 03:35-03:40 and (b) 04:00-04:05, Local Time. The shaded error bar areas correspond to the uncertainty for depolarization calculation and calibration.

➢ One case with a large difference:

[Figure]

Figure S3 The zenith-pointing MPL and zenith-pointing AVORS lidar volume depolarization ratio profiles (a) and attenuated backscatter profiles (b) on 28 May 2023, 19:55-20:00, Local Time. The shaded error bar areas correspond to the uncertainty for depolarization ratio and attenuated backscatter, respectively.

A case exhibiting a significant difference in depolarization ratio was observed on 28 May 2023, involving supercooled liquid water (~7.8 km), horizontally oriented ice crystals (HOICs, 5.8–7 km), and an aerosol layer (below 5 km), as shown in Fig. S3.

A continuous geometrically thick region of low depolarization ratio with high backscatter indicates the presence of HOICs. The depolarization ratio profiles from both lidars show good agreement, except in the regions associated with HOICs and the supercooled liquid water cloud above. The MPL lidar exhibits a lower depolarization ratio within the supercooled liquid water cloud due to its smaller field of view (FOV) and reduced multiple scattering effects. Interestingly, the MPL also shows a lower depolarization ratio in the HOIC region.

Upon examining the attenuated backscatter profiles, we found that the MPL shows a higher backscatter signal within the HOIC region, while the backscatter profiles of MPL and AVORS are nearly identical in the supercooled liquid water and other regions. This suggests that the observed depolarization difference in the HOIC region is likely due to a slight off-zenith pointing of the AVORS lidar during this case. The AVORS lidar may not be **perfectly** zenith-pointing at the time.

The MPL was located inside a vertical container and equipped with a vertical length hood to ensure strict zenith-pointing alignment (see Fig. S4). The potential maximum off-zenith angle

$\theta$ of the MPL is estimated using Eq. (1) based on the geometrical relationships illustrated in Fig. S4c and S4e.

$$\theta = \arctan\left(\frac{0.17/2}{3}\right) = 1°14' \tag{1}$$

Based on the geometrical calculation shown in Eq. (1), the potential maximum off-zenith angle of the MPL is estimated to be approximately 1° *at most*, although it is typically well aligned with the zenith direction (see Fig. S4d and S4e). In contrast, the AVORS lidar was positioned outdoors, where it is more difficult to ensure *exact 90-degree zenith-pointing*. It is therefore possible that a slight off-zenith angle existed during this observation period. This minor misalignment may explain the observed depolarization ratio discrepancies in the HOIC region.

[Figure]

[Figure]

[Figure]

Figure S4 Appearance of the MPL container (a) and the AVORS lidar system (b). (c) Schematic diagram of the MPL container. (d) Top view of the lens hood and (e) spirit level measurement at the upper edge of the MPL, demonstrating the precise zenith-pointing alignment of the MPL system.

➢   Provide the uncertainties of volume depolarization ratio profiles.  Add them in the plots.

The uncertainties in the volume depolarization ratio were generally estimated using error propagation formulas. In the figures, these uncertainties are illustrated as shaded regions. The methods used to calculate uncertainties for different systems are described below:

**Raman lidar:**

The signal of the channels is glued from the photon counting and analog signals. The uncertainties are estimated from the Poisson distribution and the Monte Carlo simulation. Subsequently, the overall uncertainty of the depolarization ratio is derived using error propagation from the individual uncertainties of each channel.

**MPL:**

The final uncertainty of MPL depolarization ratio is calculated as Eq. (2):

$$\Delta\delta_{\text{calibrated}} = \sqrt{(\Delta\delta_{\text{MPL}})^2 + (\Delta_{\text{offset}})^2} \qquad (2)$$

Here, $\Delta\delta_{\text{MPL}}$ represents the uncertainty of the uncalibrated MPL depolarization ratio, which is derived using an error propagation approach similar to that described by Heese et al. (2010). The uncertainties of the individual channels are first estimated based on Poisson statistics, and then propagated to obtain $\Delta\delta_{\text{MPL}}$.

The term $\Delta_{\text{offset}}$ refers to the uncertainty of the applied offset during calibration. It is derived from the intercept uncertainty obtained via least-squares fitting between the MPL and the reference Raman lidar depolarization ratios, as summarized in Table 1.

**AVORS lidar:**

The depolarization ratio uncertainty of AVORS lidar is calculated using Eq. (D1) from Papetta et al. (2024). Here we denote as Eq. (3).

$$(\Delta\delta)^2 = \left[\left(\frac{\Delta P^\perp}{P^\perp}\right)^2 + \left(\frac{\Delta P^\parallel}{P^\parallel}\right)^2 + \left(\frac{\Delta K^*}{K^*}\right)^2\right]\left(\frac{P^\perp}{P^\parallel K^*}\right)^2 + (\Delta g\ )^2 \qquad (3)$$

Figure S5 is the updated Fig. B1 with uncertainties in the revised manuscript.

[Figure]

Figure S5 Scatter plot of well-calibrated Raman lidar and MPL uncalibrated depolarization ratio on (a) 5 March 2022, 00:00-06:00, Local Time and (b) 14 April 2022, 00:00-05:00, Local Time; (c) Averaged Depolarization profiles on 5 March 2022, 0:00-06:00; (d) Averaged Depolarization profiles on 14 April 2022, 00:00-05:00; The horizontal gray shaded areas indicate the reference ranges used for dust and molecular layers. The shaded regions around the lines represent the uncertainties associated with depolarization ratio calculation and calibration.

➢ Discuss the "acceptable".

As shown in Fig. S5(c), Fig. S6(c), and Fig. S2, the depolarization ratio of the calibrated MPL is occasionally slightly higher than that of the AVORS lidar. However, the region below 2 km, where this difference is most pronounced, is generally unimportant for identifying horizontally oriented ice crystals (HOICs). Therefore, the imperfect performance of the calibrated depolarization ratio in this layer has minimal impact on HOIC detection.

The primary low-altitude difference is that the MPL generally exhibits higher depolarization ratios than AVORS. However, in cases of strong specular reflection, as illustrated in this study (Fig. 5b)—the MPL shows significantly lower depolarization ratios compared to the off-zenith AVORS lidar. Based on our observations, the MPL depolarization ratio can perfectly approach zero for strong backscattering targets such as liquid water clouds and HOICs. The main low-altitude bias in the MPL depolarization ratio appears in intermediate backscatter targets, such as dust layers, which are out of this study's focus.

Figures 5b and S3a demonstrate that the MPL performs well in HOIC identification, exhibiting near-zero depolarization values in HOIC-dominated regions. Importantly, the applied constant offset of 2% cannot be increased further, as doing so would result in negative depolarization ratio values in low-depolarization regions—an unphysical outcome.

Compared with previous studies such as Westbrook et al. (2010), our work represents a step forward by employing the same wavelength with depolarization measurement capabilities. Based on our experience, even when using two identical lidar systems (i.e., same model, wavelength, field of view, and detector) to observe the same target (aerosols or clouds), differences in measured depolarization ratios can still occur. These discrepancies may result from system complexity, random noise, or minor imperfections in hardware performance.

To evaluate the robustness of the separation criterion (threshold of 0.6 for the ratio of zenith to off-zenith depolarization) used in the classification flowchart (Fig. 2), we conducted a sensitivity analysis. We tested the influence of uncertainties in the depolarization ratio of the zenith and off-zenith pointing lidars using the long-term values from range bins identified as ice-containing clouds (including mixed-phase clouds and ROICs). The depolarization uncertainties range between 5-10 % (percentual error, according to Figs. S2, S5c, and S6c) for the two systems. These ranges keep the rate of falsely identified HOICs and ROICs below 2 − 5%, corroborating the tolerance of the classification scheme to uncertainties in the depolarization. For this reason, we consider the underlying uncertainties acceptable.

The discussion has been added in the revised manuscript, Line 513-515:

"From our observation, the region below 2 km, where this difference is most pronounced, is generally unimportant for identifying HOICs (in the 13 October 2022 case, HOICs are above 4 km). Therefore, the imperfect performance of the calibrated depolarization ratio in this layer has minimal impact on HOIC detection."

And add the sensitivity test result at Line 550-554:

"Sensitivity tests were performed to evaluate the stability of the 0.6 threshold (ratio of zenith to off-zenith depolarization) used in the classification flowchart (Fig. 2). Assuming typical depolarization uncertainties (5–10%, percentual error) as shown in Fig. B1, the analysis based on long-term statistics of ice-containing clouds indicates that the percentage of falsely identified HOICs and ROICs below 2-5%, which corroborates the tolerance of the classification scheme to uncertainties in the depolarization."

3. MPL lidar calibration: The volume depolarization ratio provided by the MPL lidar is compared with the corresponding measurements from the reference Raman lidar, showing an

offset in MPL data. Then, the MPL-calibrated volume depolarization values are derived from subtracting this offset. It is not clear if this offset is constant in time –in the manuscript we see that it does not change within a month. Moreover, as mentioned by the authors, the reference Raman lidar system does not operate continuously. Provide a brief description of the number of the available reference measurements used for the calibration of MPL lidar during the 1-year of measurements discussed in the manuscript (do the same for the AVORS lidar –point (3)).

**Reply:**

Available reference measurements:

The available Raman lidar data are limited to the first five months of 2022 (January to May), totaling five months. After 12 May 2022, the instrument ceased functioning due to a failure.

An ideal case for calibrating the MPL depolarization ratio involves a prolonged period of uniform dust layers coinciding with an aerosol-free region during nighttime. The presence of ice clouds above is preferable, as this study focuses on ice crystals; however, such optimal cases are rare in our observations.

For the off-zenith-pointing AVORS lidar, ice cloud cases require careful selection. Potential horizontally oriented ice crystals and heterogeneity may result in a calibrated volume depolarization ratio within the cloud that differs slightly from those of the zenith-pointing MPL or Raman lidar.

Córdoba-Jabonero et al. (2021) utilized an aerosol-free case from June 29–30, 2019, to calibrate their depolarization ratio.

In this study, we selected the cases of February 26, March 5, April 14, and May 7, 2022 (see Figs. B1 and S6), to determine a 2% offset for the MPL depolarization calibration. These same cases (February 26, March 5, April 14, and May 7, 2022) were also used to derive the parameters $K^*$ and $g$ for the AVORS lidar.

The corresponding text is added or revised to the revised manuscript, Line 507-509 and Line 536-540:

"Additional cases (not shown here) from 00:00 to 01:00 on 26 February 2022 and from 00:00 to 05:00 on 7 May 2022 further validate the consistency of the -0.02 offset."

"Three additional cases—00:00–01:00 on February 26, 2022; 00:00–05:00 on April 14, 2022; and 00:00–05:00 on May 7, 2022—were selected using a consistent method involving molecule and dust regions to derive robust $K^*$ and $g$ values. The resulting $K^*$ and g values are presented in Fig. B2 with statistics. Ultimately, we obtained $K^* = 0.962 \pm 0.006$ and $g = 0.0327 \pm 0.0009$. The calibrated depolarization ratio profile was calculated from Eq. (B3) using the robust $K^*$ and $g$."

➤ Discuss the offset of 2% is constant in time.

The last case presented in Appendix B2 is the one from 14 April 2022. To demonstrate the temporal stability of the 2% depolarization ratio offset, we selected the earliest available Raman lidar observation from February 2022 (data quality in January was insufficient) and the

latest available case from May 2022. Two additional cases used in the depolarization calibration (26 February and 7 May 2022) are shown in Fig. S6. As shown in Fig. S6 and Table 1, the 2% offset remains consistent over time. After May 2022, Raman lidar data became unavailable. However, the MPL system was not physically moved after that time, and the internal temperature of the MPL container was kept stable using air conditioning. Furthermore, the calibrated MPL and AVORS lidars showed excellent agreement on 21 May 2023, over one year later. Based on these factors, we confidently assume that the 2% offset remains valid after May 2022.

In addition, we tested the two-parameter calibration method proposed by Papetta et al. (2024), which introduces an additional factor $K*$ to improve calibration accuracy. While this method yielded near-perfect results for the calibration case, it exhibited signs of overfitting and did not perform well for other cases. Consequently, we opted to retain the simpler, constant 2% offset approach.

It is worth noting that the depolarization ratio difference between the MPL and the reference Raman lidar was not entirely uniform across range. A slightly larger discrepancy was observed below 2 km, where the MPL tended to report higher depolarization ratios. We attribute this either to differences in the overlap of the polarization channels in the reference Raman system, or to near-range effects in the MPL, such as the deadtime effect. However, this height range is not critical for our study, as HOIC (horizontally oriented ice crystals) occurrences below 2 km are extremely rare—in this study, all were observed above 4 km.

Importantly, we could not increase the offset beyond 2%, as the molecular scattering region already yields a depolarization ratio close to 0.0004. A larger offset would result in negative depolarization values in this region, which is physically implausible. Additionally, the specular reflection region in Fig. 5b shows depolarization values only slightly above zero; applying a larger offset would drive these values into the negative, which would again be unphysical. For these reasons, we maintained the 2% offset throughout the study.

[Figure]

Figure S6 Scatter plot of well-calibrated Raman lidar and MPL uncalibrated depolarization ratio on (a) 26 February 2022, 00:00-01:00, Local Time and (b) 07 May 2022, 00:00-05:00, Local Time; (c) Averaged Depolarization profiles on 26 February 2022, 0:00-01:00; (d) Averaged Depolarization profiles on 07 May 2022, 00:00-05:00; The horizontal grey shade regions indicate the reference ranges used for dust and molecular layers. The shaded error bar areas correspond to the uncertainty for depolarization calculation and calibration.

The 2% offset, derived from the intercept of the least-squares fit, is based on precise values reported in Table 1, yielding a final average offset of -0.02011 ± 0.00101."

Table 1. Average intercept of the least-squares fit for each examined period with the standard deviations.

| Date | 20220226 | 20220305 | 20220414 | 20220507 | average |
|---|---|---|---|---|---|
| offset | -0.02030 ±0.00051 | -0.02015 ± 0.00006 | -0.02003 ±0.00220 | -0.01999 ±0.00336 | -0.02011 ±0.00101 |

4. AVORS lidar calibration: Provide brief statistics on K* and g, for all reference measurements (from Raman lidar) used to characterize the AVORS lidar depolarization calibration, similar to what is presented in Fig. 7 of Papetta et al. (2024).

**Reply:**

Thank you for pointing this out! We evaluated four different cases, resulting in updated values of $K*$ and $g$, as presented in Table 2 and Fig. S7. The values of $K*$ and $g$ were revised from 0.954 and 0.0329 to 0.962 and 0.0327, respectively. All figures related to depolarization were redrawn using these updated parameters. The calibrated depolarization ratio exhibits only minor changes with the revised $K*$ and $g$.

Figure S7 has been added to the manuscript as new Fig. B2.

The derived $K*$ and $g$ values are relatively stable compared to the highly variable cases presented in Fig. 7 of Papetta et al. (2024). In their study, the Cimel lidar under calibration and the reference Raman lidar were separated by tens of kilometers rather than co-located, which likely contributes to the greater variability observed in $K*$ and $g$.

Table 2. Average polarization parameters for each examined period are given with the standard deviation.

| Date | 20220226 | 20220305 | 20220414 | 20220507 | average |
|---|---|---|---|---|---|
| *K** | 0.9649 ± 0.0036 | 0.9545 ± 0.0029 | 0.9675 ± 0.0045 | 0.9618 ± 0.0043 | 0.9622 ± 0.0062 |
| *g* | 0.0312 ± 0.0002 | 0.0329 ± 0.0002 | 0.0330 ± 0.0002 | 0.0337 ± 0.0003 | 0.0327 ± 0.0009 |

[Figure]

Figure S7 Temporal evolution of polarization parameters derived using the two-parameter approach. Error bars indicate the variability of the derived parameters within the selected molecular scattering and dust reference layers. The average polarization parameter value and its standard deviation in the whole period is shown by solid green lines and dashed gray lines, respectively. The timestamps of the cases shown in Fig. B1 are highlighted in red.

Other revisions:

Line 68: Replace "Passive satellite" with "Passive satellites".

**Reply:** Done!

Line 206: Replace "If by contrast, the above δoff-zenith > 0.1 ice-containing cloud pixels show δzenith < 0.1…" with "If the cloud parts categorized as ROICs or MCPs have δoff-zenith > 0.1 and δzenith < 0.1…".

**Reply:** Done!

Line 208: Replace "…i.e…." with "…since…."

**Reply:** Done!

Lines 209-210: Replace "…kept their ROIC or MPC labels." with "categorized as ROIC or MPC."

**Reply:** Done!

Line 214: "The threshold values were fixed empirically from the whole cloud dataset collected during 2022." And lines 226-230 "For the identification scheme, … for specular reflection." Support these empirically-derived values with values provided in the literature from lidar measurements and scattering calculations of ROICs and HOICs.

**Reply:** I have added some literature as follows.

0.1 and 0.3 are frequently used criteria for liquid droplet and ice crystal discrimination. In principle, liquid water droplets show a low depolarization ratio, and ice crystals show a high depolarization ratio. The criteria of 0.3 are quite common for ice crystal identification. Due to the strong multiple scattering effect from liquid water clouds, the criterion of liquid water droplets is not exactly zero but slightly larger than zero. Since different fields of view contribute to different scales of multiple scattering. Here, the criteria of 0.1 are determined from long-term observation (> 0℃).

2 and 0.6 are simply statistical results from the long-term yearlong dataset (see Fig. 3b). We assume that most of the cloud range bins are ROIC-dominant and not strongly affected by the HOICs. And the upper left branch of Fig. 3b is are HOIC dominant region. They are new criteria, so we do not have literature to support them.

The relevant literature has been added in the revise manuscript Line 220-221:

"(Seifert 2011; Lewis et al. 2020; Whitehead et al., 2024)"

Lines 330-334: "Figure 7 shows … where HOICs exist." Provide possible reasons why this happens. Is there a possibility that the results shown in Fig. 7 are biased due to the attenuation of the lidar signals, which may result to non-identification of HOICs at higher heights and lower temperatures? Discuss.

**Reply:** Thank you for pointing out this! We added sentences to try to explain the results, Line 339-340 and Line 343-344:

"Higher temperatures ($-8 > T > -22 \, °C$) favor the formation of plate-like ice crystals, while weaker horizontal winds exert less torque to disturb their quasi-horizontal orientation."

"Overall, in the troposphere, higher altitudes are associated with lower temperatures and stronger horizontal winds—a pattern that also applies to regions where HOICs are present."

We added a discussion part after the sentence, Line 344-349. Ground-based lidar-based cloud research always has this defect from lidar attenuation.

"It should be noted that the non-detection of HOICs at higher altitudes due to lidar signal attenuation may introduce a slight bias in the results, potentially leading to an overestimation of temperature and an underestimation of horizontal wind speed. This limitation is common in ground-based lidar studies of clouds. However, based on radar observations indicating a cloud top height of approximately 7 km and radiosonde data showing a minimum temperature above $-22 \, °C$ and maximum horizontal wind speeds below $25 \, \mathrm{ms}^{-1}$, the overall bias is expected to be minor. Therefore, the main conclusions of this study remain robust."

Line 336: Replace "…spectral with…" with "…spectral width…".

**Reply:** Done!

Lines 349-350: "Contrary … as shown in Fig. 4(l)." Provide possible reasons.

**Reply:** The LDR of plate-like ice crystals is too small. The current sensitivity of cloud radar cannot detect it. We have added, Line 365-366:

"This is likely because the LDR of plate-like ice crystals is too small for the current sensitivity of cloud radar to detect."

Line 434: Replace "…are require help…" with "….are required to help…"

**Reply:** Done!

Lines 457-459: Replace "The photo count rate … laser energy." with corresponding equation.

**Reply:** Thank you for pointing this out! I added the corresponding equation after the sentence:

"The above steps are summarized as the Eqs. B1 and B2:

$$NRB_{MPL} = \frac{[(Photo\ Count\ Rate \times Dead\ Time\ Correction) - Afterpluse - Background] \times range^2}{Overlap \times Laser\ Energy} \quad (B1)$$

$$NRB_{AVORS} = \frac{[(Photo\ Count\ Rate \times Dead\ Time\ Correction) - Background] \times range^2}{Overlap \times Laser\ Energy} \quad (B2)$$

Line 498: Replace "…Fig. 2c." with "…Fig. B1c."

**Reply:** Done!

References:

Córdoba-Jabonero, C., Ansmann, A., Jiménez, C., Baars, H., López-Cayuela, M.-Á., and Engelmann, R.: Experimental assessment of a micro-pulse lidar system in comparison with reference lidar measurements for aerosol optical properties retrieval, Atmospheric Measurement Techniques, 14, 5225–5239, https://doi.org/10.5194/amt-14-5225-2021, 2021.

Freudenthaler, V.: About the effects of polarising optics on lidar signals and the Δ90 calibration, Atmos. Meas. Tech., 9, 4181–4255, https://doi.org/10.5194/amt-9-4181-2016, 2016.

Heese, B., Flentje, H., Althausen, D., Ansmann, A., and Frey, S.: Ceilometer lidar comparison: backscatter coefficient retrieval and signal-to-noise ratio determination, Atmos. Meas. Tech., 3, 1763–1770, https://doi.org/10.5194/amt-3-1763-2010, 2010.

Lewis, J. R., Campbell, J. R., Stewart, S. A., Tan, I., Welton, E. J., and Lolli, S.: Determining cloud thermodynamic phase from the polarized Micro Pulse Lidar, Atmos. Meas. Tech., 13, 6901–6913, https://doi.org/10.5194/amt-13-6901-2020, 2020.

Papetta, A., Marenco, F., Kezoudi, M., Mamouri, R.-E., Nisantzi, A., Baars, H., Popovici, I. E., Goloub, P., Victori, S., and Sciare, J.: Lidar depolarization characterization using a reference system, Atmospheric Measurement Techniques, 17, 1721–1738, https://doi.org/10.5194/amt-17-1721-2024, 2024

Seifert, P.: Dust-related ice formation in the troposphere: A statistical analysis based on 11 years of lidar observations of aerosols and clouds over Leipzig, PhD thesis, University of Leipzig, Leipzig, Germany, https://nbn-resolving.org/urn:nbn:de:bsz:15-qucosa-71167, 2011

Wang, N., Zhang, K., Shen, X., Wang, Y., Li, J., Li, C., ... & Liu, D. Dual-field-of-view high-spectral-resolution lidar: Simultaneous profiling of aerosol and water cloud to study aerosol–cloud interaction. Proceedings of the National Academy of Sciences, 119(10), e2110756119. https://doi.org/10.1073/pnas.2110756119, 2022

Westbrook, C., Illingworth, A., O'Connor, E., and Hogan, R.: Doppler lidar measurements of oriented planar ice crystals falling from supercooled and glaciated layer clouds, Quarterly Journal of the Royal Meteorological Society, 136, 260–276, https://doi.org/10.1002/qj.528, 2010.

Whitehead, L. E., McDonald, A. J., and Guyot, A.: Supercooled liquid water cloud classification using lidar backscatter peak properties, Atmos. Meas. Tech., 17, 5765–5784, https://doi.org/10.5194/amt-17-5765-2024, 2024.